# Selective eradication of cancer displaying hyperactive Akt by exploiting the metabolic consequences of Akt activation

**Veronique Nogueira[1]\*, Krushna C Patra[1†], Nissim Hay[1,2]\***

[1]Department of Biochemistry and Molecular Genetics, College of Medicine, University of Illinois at Chicago, Chicago, United States; [2]Research & Development Section, Jesse Brown VA Medical Center, Chicago, United States

**Abstract** Akt activation in human cancers exerts chemoresistance, but pan-Akt inhibition elicits adverse consequences. We exploited the consequences of Akt-mediated mitochondrial and glucose metabolism to selectively eradicate and evade chemoresistance of prostate cancer displaying hyperactive Akt. PTEN-deficient prostate cancer cells that display hyperactivated Akt have high intracellular reactive oxygen species (ROS) levels, in part, because of Akt-dependent increase of oxidative phosphorylation. High intracellular ROS levels selectively sensitize cells displaying hyperactive Akt to ROS-induced cell death enabling a therapeutic strategy combining a ROS inducer and rapamycin in PTEN-deficient prostate tumors in mouse models. This strategy elicited tumor regression, and markedly increased survival even after the treatment was stopped. By contrast, exposure to antioxidant increased prostate tumor progression. To increase glucose metabolism, Akt activation phosphorylated HK2 and induced its expression. Indeed, HK2 deficiency in mouse models of Pten-deficient prostate cancer elicited a marked inhibition of tumor development and extended lifespan.

DOI: https://doi.org/10.7554/eLife.32213.001

\*For correspondence:
vnogueir@uic.edu (VN);
nhay@uic.edu (NH)

**Present address:** †Massachusetts General Hospital Cancer Center, Harvard Medical School, Boston, United States

**Competing interests:** The authors declare that no competing interests exist.

## Introduction

One of the most frequent events in human cancer is hyperactivation of the serine/threonine kinase Akt. Akt is hyperactivated in cancer by multiple mechanisms, largely through the activation of its upstream regulator phosphoinositide 3-kinase (PI3K), which generates the phosphatidylinositol-3,4,5-trisphophate ($PIP_3$) required for Akt activation (*Mayer and Arteaga, 2016*). The activity of PI3K is negatively regulated by the tumor suppressor Phosphatase And Tensin Homolog (PTEN), which is a PIP3 phosphatase, and therefore inhibits the PI3K/Akt signaling pathway. PTEN expression is often lost in human cancers, specifically in glioblastoma, melanoma, endometrial, and prostate cancers (*Hollander et al., 2011*). The activation of PI3K/Akt signaling in cancer and its ability to exert chemoresistance led to the development of small molecule inhibitors of PI3K and Akt, which are currently being tested in clinical trials (*Kim et al., 2005*; *Zhang et al., 2017*; *Zheng, 2017*). There are three Akt genes in mammalian cells (*Akt1-3*), and their encoded proteins have a high degree of identical amino acids; however, the expression pattern in mammalian tissues and organs is different among the three isoforms. While Akt1 is ubiquitously expressed, Akt2 is expressed at the highest level in insulin-responsive tissues, and Akt3 is expressed at the highest level in the brain. Different mouse phenotypes with individual Akt isoform germ line deletions can be explained by their relative expression in the organs that determine the phenotype (*Dummler and Hemmings, 2007*; *Hay, 2011*). The Akt inhibitors currently in clinical trials are pan-Akt inhibitors that inhibit the different Akt isoforms to a similar extent; however, these pan-Akt inhibitors exert undesired side effects, such as hyperglycemia, hyperinsulinemia, and diabetes (*Wang et al., 2017*). Furthermore, genetic

deletion of Akt1 and Akt2 in the mouse liver induces liver damage, inflammation, and paradoxically hepatocellular carcinoma (HCC) (*Wang et al., 2016*). Development of isoform-specific inhibitors is one option in attempts to reduce the undesired systemic consequences of pan-Akt inhibition, although this is challenging. Alternatively, a therapeutic approach that selectively targets cancer cells displaying hyperactive Akt is desirable.

Perhaps the most evolutionarily conserved function of Akt is mediating cellular and organismal metabolism. It seems likely that this conserved function of Akt is lused by cancer cells to fulfill their anabolic demands. As PTEN is lost in approximately 40% of prostate cancers (*Pourmand et al., 2007*; *Taylor et al., 2010*), we chose to work towards developing a personalized therapeutic approach using PTEN-deficient prostate cancer to explore selective vulnerability as a consequence of Akt's metabolic activity. As we showed previously, activation of Akt increases both glycolysis and oxidative phosphorylation (*Gottlob et al., 2001*; *Robey and Hay, 2009*). We also showed that Akt activation increases intracellular ROS levels, in part by increasing oxidative phosphorylation. As Akt does not exert resistance to ROS-induced cell death, increasing ROS levels could selectively eradicate cells displaying hyperactive Akt (*Nogueira et al., 2008*). Here, we showed that human PTEN-deficient and not PTEN-proficient prostate cancer cells have high intracellular ROS levels, which are Akt-dependent. The high level of ROS can be exploited to selectively eradicate human PTEN-deficient tumors in vivo as well as in a mouse model of Pten-deficient prostate cancer. We used the natural compound phenylethyl isothiocyanate (PEITC), which depletes intracellular glutathione (*Xu and Thornalley, 2001*), as a ROS inducer either alone or in combination with the mTORC1 inhibitor rapamycin to selectively eradicate Pten-deficient cancer cells in vivo. We also found that in PTEN-deficient prostate cancer, HK2 is induced because of Akt activation to increase glycolysis. HK2 is the hexokinase isoform that is not highly expressed in most mammalian tissues but is generally induced in cancer cells by multiple mechanisms (*Hay, 2016*). Furthermore, HK2 is phosphorylated by Akt to increase its mitochondrial binding (*Miyamoto et al., 2008*) and therefore its glycolytic activity (*DeWaal et al., 2018*). Here we show that silencing HK2 in human PTEN-deficient prostate tumors and deleting HK2 in a mouse model of Pten-deficient prostate cancer inhibits cancer development in both cytostatic and cytotoxic manners. HK2 deficiency also overcomes the chemoresistance of PTEN-deficient prostate cancer cells.

## Results

### PTEN-deficient human prostate cancer cells display high oxygen consumption, OXPHO, and high levels of ROS

We employed three human prostate cancer cell lines: DU145, which is PTEN-proficient, and PC3 and LNCaP, which are PTEN-deficient. As expected, the PTEN status in DU145, PC3, and LNCaP cells determines Akt activity in these cells (*Figure 1A*). DU145 cells, which harbor wild-type PTEN, exhibit low Akt activity. PTEN-deficient PC3 and LNCaP cells display hyperactivated Akt. As we previously found, Akt elevates oxygen consumption and intracellular ROS levels (*Nogueira et al., 2008*). We therefore determined these two parameters in prostate cancer (CaP) cells in which PTEN is often lost. Basal oxygen consumption was lowest in the PTEN-proficient DU145 cells, while it was gradually increased in the PTEN-deficient PC3 and LNCaP cells (*Figure 1B*), following the pattern of Akt activity in which higher oxygen consumption was correlated with higher Akt activity. Silencing Akt1 and Akt2 in PC3 cells markedly decreased oxygen consumption, indicating that the high oxygen consumption in these cells is Akt-dependent (*Figure 1C*). Interestingly, basal oxygen consumption in DU145 cells reached the maximum capacity of the respiratory chain, while PC3 and LNCaP cells had a larger spare capacity. *Figure 1B* also shows that the ATP production capacity is two-fold higher in PC3 and LNCaP cells compared with DU145 cells, agreeing with our previous observations that Akt activation increases ATP production by both glycolysis and oxidative phosphorylation (*Gottlob et al., 2001*). As intracellular ROS are by-products of high OXPHO, we determined intracellular ROS production at the cytosolic (*Figure 1D*) and mitochondrial (*Figure 1E*) levels, and found that high Akt activity was correlated with high intracellular levels of ROS. Akt1 and Akt2 knockdown in PC3 cells consistently decreased ROS levels, confirming that Akt regulates intracellular ROS levels. (*Figure 1F*). Finally, we found that in PC3 and LNCaP cells, mitochondrial membrane potential is

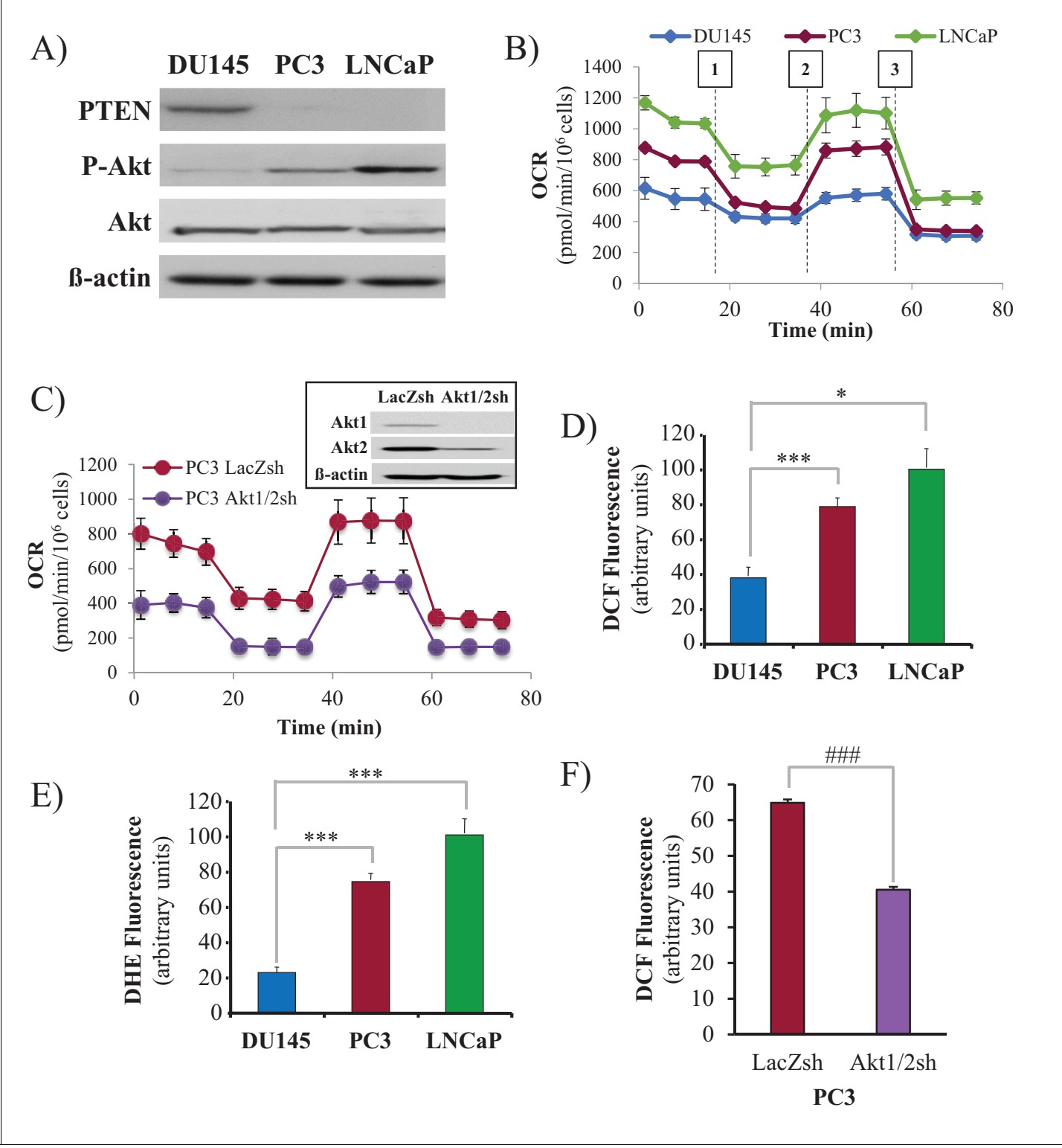

**Figure 1.** Akt activation in PTEN-deficient prostate cancer cells elevates oxygen consumption and intracellular ROS levels. The human CaP cells DU145, PC3, and LNCaP were seeded in 10% FBS and harvested after two days to measure various parameters. (A) Immunoblot showing the expression levels of PTEN, P-Akt (ser 473), pan-Akt, and ß-actin as a loading control. (B) Oxygen consumption: OCR was measured using the Seahorse XF96$^e$ analyzer for all three CaP cell lines. After the OCR was established, oligomycin (1), FCCP (2), and rotenone/antimycin A (3) were added sequentially. The traces shown are representative of three independent experiments in which each data point represents technical replicates of four wells each ± SEM. (D, E) Relative ROS levels: CaP cells were incubated with H2DCFDA (D) or DHE (E), and the levels of fluorescence were analyzed by flow cytometry as an

*Figure 1 continued on next page*

*Figure 1 continued*

indicator of ROS levels. Data represent the mean ±SEM of three independent experiments performed in triplicate. *p<0.01, ***p<0.005 versus DU145. No significant differences between PC3 and LNCaP were observed. (C, F) Akt1 and Akt2 were knocked down in PC3 cells, and the OCR (C) and cytosolic ROS levels (F) were measured. The results are presented as the average of at least three independent experiments performed in triplicate ± SEM. ###p<0.0001 versus PC3 LacZsh. Insert in (C) shows the expression levels of Akt1, Akt2, and ß actin as a loading control in PC3 cells in which Akt1 and Akt2 were knocked down.

DOI: https://doi.org/10.7554/eLife.32213.002

The following figure supplements are available for figure 1:

**Figure supplement 1.** Mitochondrial membrane potential measured as JC-1 aggregate to monomer ratio.

DOI: https://doi.org/10.7554/eLife.32213.003

**Figure supplement 2.** Immunoblot showing the expression levels of the detoxifying enzymes catalase, MnSOD, and Cu/ZnSOD (ß actin as a loading control) in all three CaP cell lines.

DOI: https://doi.org/10.7554/eLife.32213.004

**Figure supplement 3.** Level of Sesn3 mRNA relative to that of actin in CaP cells, as assessed by quantitative RT-PCR.

DOI: https://doi.org/10.7554/eLife.32213.005

**Figure supplement 4.** Immunoblot showing the expression levels of sestrin 3 (SESN3) and ß actin as a loading control.

DOI: https://doi.org/10.7554/eLife.32213.006

**Figure supplement 5.** Level of ROS, as assessed by flow cytometry, after incubation with H2DCFDA.

DOI: https://doi.org/10.7554/eLife.32213.007

higher than in DU145 cells (*Figure 1—figure supplement 1*), which is likely correlated with the higher respiratory chain activity in PC3 and LNCaP cells.

In our previous studies, we found that Akt activation increases ROS not only by increasing oxygen consumption but also by inhibiting the expression of ROS scavengers downstream of FOXO, such as MnSOD and catalase, and particularly sestrin3 (Sesn3) (*Nogueira et al., 2008*). Sesn3 is a transcriptional target of FOXO (*Chen et al., 2010*) and a member of a protein family including Sesn1 and Sesn2, which reduce ROS by several mechanisms (*Bae et al., 2013*; *Kopnin et al., 2007*). Interestingly, in contrast to our findings in MEFs (*Nogueira et al., 2008*), changes in MnSOD and catalase expression in the CaP cells did not correlate with changes in ROS levels (*Figure 1—figure supplement 2*), which is consistent with previously observations (*Chowdhury et al., 2007*). However, the pattern of Sesn3 expression was consistent with ROS levels, and while DU145 cells express high levels of Sesn3, PC3 and LNCaP cells express relatively low levels of Sesn3 (*Figure 1—figure supplement 3*). Interestingly, downregulation of Sesn3 in DU145 cells or up-regulation of Sesn3 in PC3 cells (*Figure 1—figure supplement 4*) was sufficient to modulate cytosolic ROS production in these cells (*Figure 1—figure supplement 5*). Sesn3 knockdown in DU145 cells increased ROS production, while overexpression of Sesn3 in PC3 cells decreased ROS production. These results suggest that Sesn3 contributes to the regulation of intracellular ROS downstream of Akt and FoxOs in CaP cells.

Taken together, these results show that PTEN-deficient prostate cancer cells display high OXPHO and ROS levels in an Akt-dependent manner.

## PTEN-deficient prostate cancer cells are selectively sensitized to killing by a ROS inducer

We previously reported that cells that display high Akt activity could be selectivity killed by increasing the intracellular level of ROS (*Nogueira et al., 2008*). This selectivity results from the high intracellular ROS levels exerted by Akt activation in combination with the inability of Akt to protect against ROS-induced cell death. We therefore treated the prostate cancer cells with 2-methoxyestradiol (2-ME), an endogenous metabolite of estradiol-17β that increases ROS, or with β-phenylethyl isothiocyanate (PEITC), a natural compound found in consumable cruciferous vegetables that is known to increase intracellular ROS levels by depleting intracellular glutathione (*Ting et al., 2010*; *Yu et al., 1998*; see also *Figure 2—figure supplement 1*). We found that CaP cells with high Akt activity resulting from the loss of PTEN (LNCaP, PC3 cells) were more vulnerable to 2-ME- and PEITC-induced cell death than PTEN-proficient CaP cells (DU145 cells) (*Figure 2A,B*, and *Figure 2—figure supplement 2*). LNCaP and PC3 cells are more vulnerable to the glutathione reducing agent BSO (*Figure 2—figure supplement 3*). Interestingly, the NADP+/NADPH ratio is elevated in the PTEN-deficient cells (*Figure 2—figure supplement 4*). The elevated NADP+/NADPH could be

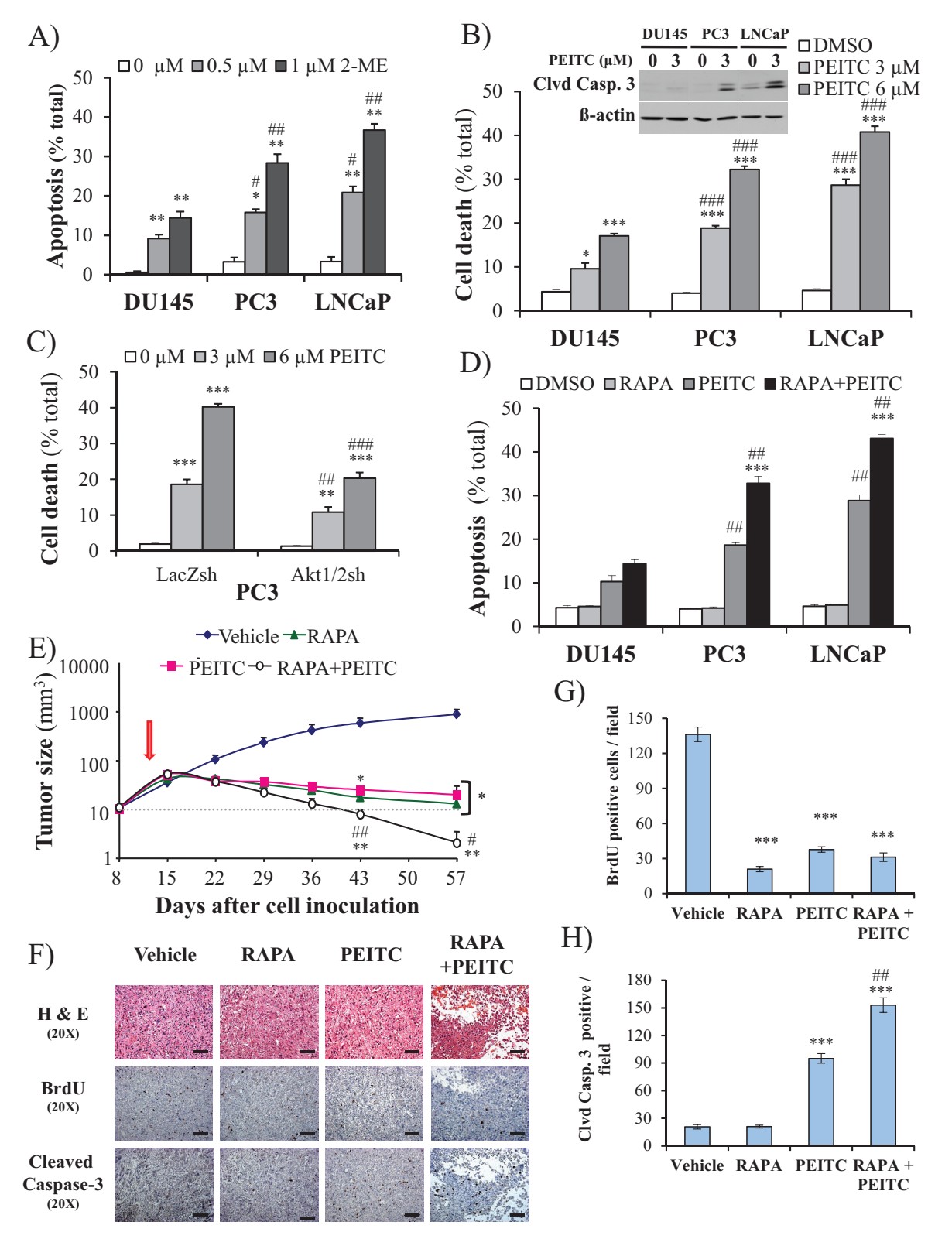

**Figure 2.** ROS inducers and the combination of a ROS inducer and rapamycin induce CaP PTEN-deficient cell death in vitro and eradicate their tumors in vivo. (A) CaP cell lines were incubated with 2-ME for 24 hr, the cells were fixed and apoptosis was quantified by DAPI staining. The data represent the mean ±SEM of three independent experiments performed in triplicate. *p<0.005, **p<0.002 versus DMSO (0 μM) for each cell line. #p<0.02, ##p<0.01 versus DU145. (B) CaP cell lines were incubated with PEITC, collected and fixed for estimation of cell death by PI staining or lysed to extract

*Figure 2 continued on next page*

*Figure 2 continued*

total protein. They were then subjected to immunoblotting with cleaved caspase-3 and ß-actin as a loading control (insert). The data represent the mean ±SEM of three independent experiments performed in triplicate. *p<0.005, ***p<0.001 versus DMSO for each cell line. ###p<0.0005 versus DU145. (C) PC3 Akt1/2 knockdown cells were incubated with PEITC for 17 hr, and then cell death was estimated by PI staining as the percentage of apoptotic cells among total cells. The data represent the mean ±SEM of three independent experiments performed in triplicate. **p<0.001, ***p<0.0001 versus DMSO for each cell line. ##p<0.005, ###p<0.0001 versus PC3 LacZsh. (D) CaP cells were incubated for 8 hr with 20 nM rapamycin (RAPA) prior to the addition of PEITC (3 µM). After 17 hr of incubation with PEITC, the cells were fixed, and apoptosis was quantified by DAPI staining. The data represent the mean ±SEM of three independent experiments performed in triplicate. ***p<0.0001 versus PEITC for each cell line. ##p<0.0005 versus DU145. (E–H) In vivo therapeutic effect of rapamycin +PEITC in mice inoculated with PC3 prostate cancer cells. Thirty-two nude mice were subcutaneously injected with PC3 cells in both flanks and randomly divided into four groups (eight mice per group, 16 tumors per group) for treatment with PEITC, rapamycin (RAPA), a combination of RAPA +PEITC, or a solvent control (Vehicle). (E) Graph presenting the tumor growth rates in each group. Treatment began on day 13 (~15 mm$^3$, red arrow) and stopped on day 43 after tumor cell inoculation. The data represent the average size ±SEM of 16 tumors up to day 43. The data collection from day 57 represent the average size of the eight remaining xenograft tumors only. *p<0.003, **p<0.002 versus vehicle. #p<0.03, ##p<0.01 versus PEITC or RAPA. (F) Cross-sections of tumors collected from the experiment described in (E). At day 50 after tumor cell inoculation, the tumor cross-sections were subjected to hematoxylin and eosin (H and E, top) staining, BrdU staining (middle), and anti-cleaved caspase-3 staining (bottom). Scale bars: 100 µm. (G, H) Histograms showing quantification of the positively stained cells in (F). The results are presented as the mean ±SEM of the positively stained cells of four sections from four treated mice. The stained cells were counted in four random fields of each section. ***p<0.0002 versus vehicle. ##p<0.001 versus PEITC.

DOI: https://doi.org/10.7554/eLife.32213.008

The following figure supplements are available for figure 2:

**Figure supplement 1.** Glutathione levels (Left) and GSH/GSSG ratio (Right) in CaP cells after 8 hr incubation with DMSO or PEITC 6 µM.
DOI: https://doi.org/10.7554/eLife.32213.009

**Figure supplement 2.** (Left) Apoptosis was measured on live cells by caspase 3/7 activity assay after drug treatment: 2-ME 1 µM (14 hr) or 20 nM rapamycin (5 hr) followed by 6 µM PEITC (8 hr).
DOI: https://doi.org/10.7554/eLife.32213.010

**Figure supplement 3.** CaP cell lines were incubated with BSO (2 mM) for 36 and 42 hr, the cells were fixed and cell death was quantified by PI staining.
DOI: https://doi.org/10.7554/eLife.32213.011

**Figure supplement 4.** NADP$^+$/NADPH ratio in CaP cells.
DOI: https://doi.org/10.7554/eLife.32213.012

**Figure supplement 5.** After modulation of SESN3 expression, PC3 and DU145 cells were treated with PEITC (0, 3 and 6 µM) for 17 hr, the cells were fixed and cell death was assessed by DAPI staining.
DOI: https://doi.org/10.7554/eLife.32213.013

**Figure supplement 6.** DU145, PC3, and LNCaP cells were incubated with N-acetylcysteine (100 µM NAC) for 2 hr prior to 17 hr of incubation with PEITC (6 µM) in the presence of NAC or not.
DOI: https://doi.org/10.7554/eLife.32213.014

**Figure supplement 7.** Immunoblot showing the expression of PTEN (and HA-Tag), and ß-actin as a loading control after PTEN was downregulated in DU145 cells (1: control shLacZ, 2: shPTEN) or overexpressed in PC3 and LNCaP cells (3: control pBP, 4: pBP-PTEN).
DOI: https://doi.org/10.7554/eLife.32213.015

**Figure supplement 8.** PTEN expression determines the levels of ROS and oxygen consumption.
DOI: https://doi.org/10.7554/eLife.32213.016

**Figure supplement 9.** Cells were incubated with PEITC or rapamycin/PEITC for 17 hr and scored for apoptosis 17 hr later by DAPI staining.
DOI: https://doi.org/10.7554/eLife.32213.017

**Figure supplement 10.** mAkt was stably overexpressed in DU145.
DOI: https://doi.org/10.7554/eLife.32213.018

**Figure supplement 11.** ROS levels and ROS-induced cell death are Akt-dependent.
DOI: https://doi.org/10.7554/eLife.32213.019

**Figure supplement 12.** Rapamycin elevates Akt activity.
DOI: https://doi.org/10.7554/eLife.32213.020

**Figure supplement 13.** Rapamycin increases the ROS levels induced by PEITC.
DOI: https://doi.org/10.7554/eLife.32213.021

**Figure supplement 14.** Torin, not rapamycin, decreases the OCR and ROS levels in PTEN-deficient CaP cells.
DOI: https://doi.org/10.7554/eLife.32213.022

**Figure supplement 15.** In vivo therapeutic effects of rapamycin +PEITC in mice inoculated with DU145 prostate cancer cells.
DOI: https://doi.org/10.7554/eLife.32213.023

either contributing to the high level of ROS or it is a result of increased NADPH consumption to combat the high ROS level. Alternatively or additionally, an increase in NADPH consumption for fatty acid synthesis in the PTEN-deficient cells can contribute to the higher NADP+/NADPH ratio.

Silencing *Sesn3* increased PEITC-induced cell death in DU145 cells, and overexpression of SES-N3in PC3 cells decreased their sensitivity to PEITC (*Figure 2—figure supplement 5*). The cell death induced by PEITC is ROS-dependent as it is inhibited by the ROS scavenger N-acetyl cysteine (NAC) (*Figure 2—figure supplement 6*). To determine whether the hypersensitivity of PTEN-deficient prostate cancer cells to ROS-induced cell death is PI3K/Akt dependent, we first restored PTEN expression in the Pten-deficient cells and silenced *Pten* in the Pten-proficient cells. (*Figure 2—figure supplement 7*). Oxygen consumption and ROS production were increased by silencing *Pten* in DU145 cells and decreased in PC3 and LNCaP cells expressing PTEN (*Figure 2—figure supplement 8*). The silencing of *Pten* in DU145 cells increased sensitivity to PEITC, whereas the expression of PTEN in PC3 and LNCaP cells decreased their sensitivity to PEITC (*Figure 2—figure supplement 9*). Like the silencing of PTEN in DU145 cells, expression of activated myristoylated Akt (mAkt) in DU145 cells increased ROS levels and renders the cells more sensitive to ROS-induced cell death (*Figure 2—figure supplement 10*). Finally, the knockdown of *Akt1* and *Akt2* in PC3 and LNCaP cells that reduced ROS levels also rendered them resistant to PEITC-induced cell death (*Figure 1F*, *Figure 2C*, and *Figure 2—figure supplement 11*). We concluded that Akt activation in Pten-deficient prostate cancer cells could not protect against oxidative stress-induced cell death, but rather sensitized the cells to ROS-induced cell death by increasing their intracellular ROS levels.

## Treatment with PEITC and rapamycin inhibits and regresses tumor development in a xenograft model and in a mouse model of prostate cancer

We previously showed that rapamycin treatment could further sensitize cells displaying hyperactive Akt to oxidative stress-induced cell death, which could result, in part, from the further activation of Akt by inhibition of mTORC1 inhibitory activity on the PI3K/Akt signaling (*Nogueira et al., 2008*). This was also observed in prostate cancer cells (*Figure 2—figure supplement 12*). Thus, a combination of rapamycin and oxidative stress could not only circumvent resistance to cell death but also selectively kill cells treated with rapamycin. Before applying this strategy to animal models of prostate cancer, we first established our proof-of-concept with prostate cancer cells in vitro. As shown in *Figure 2D*, rapamycin alone did not induce cell death, but pretreatment with rapamycin augmented the ability of PEITC to induce cell death in all three CaP cell lines. Although rapamycin treatment increased PEITC-induced cell death in all cell lines, the LNCaP and PC3 cells with hyperactivated Akt were markedly more sensitive to cell death induced by the combination of rapamycin and PEITC than DU145 cells (*Figure 2D*). The synergistic effect of rapamycin and PEITC on cell death could be explained by the induction of ROS exceeding the scavenging capacity (*Figure 2—figure supplement 13*). We found that rapamycin, by itself, does not substantially affect oxygen consumption or intracellular ROS induced by Akt (*Figure 2—figure supplement 14*). This contrasts with the catalytic inhibitor of mTOR, torin1, which decreased oxygen consumption and ROS levels (*Figure 2—figure supplement 14*). These results are consistent with previously published results showing that while the mTOR kinase inhibitor inhibits OXPHO in an eIF4E-dependent manner, rapamycin does not (*Morita et al., 2013*). We concluded that a combination of rapamycin and PEITC could be used to selectively kill prostate cancer cells expressing hyperactive Akt.

To examine the efficacy of the strategy to selectively eradicate prostate cancer cells carrying activated Akt in vivo, we first employed xenografts of PC3 cells in athymic nude mice and studied the effect of PEITC and rapamycin on the growth of tumors induced by PC3 cells (*Figure 2E*). After tumor onset, the mice were either not treated or treated with rapamycin alone, PEITC alone, or a combination of both rapamycin and PEITC. Rapamycin alone or PEITC alone significantly attenuated the growth of the tumors, but the tumors remained palpable. However, the combination of PEITC and rapamycin regressed tumor growth and eradicated the tumors. Analyses of tumor sections near the endpoint of the experiment showed that PEITC alone induced both a profound inhibition of BrdU incorporation and cell death, as assessed by cleaved caspase 3, whereas rapamycin alone did not induce cell death but did inhibit BrdU incorporation (*Figure 2F–H*). Cell death after treatment with both PEITC and rapamycin, as measured by cleaved caspase 3, was profoundly higher than that induced by PEITC alone (*Figure 2F–H*). When the PTEN-proficient DU145 xenografts were similarly

treated, the effect of rapamycin alone or PEITC alone on tumor growth was not as profound (*Figure 2—figure supplement 15*). Importantly, the combination of rapamycin and PEITC did not decrease tumor growth as it did for the PTEN-deficient PC3 xenografts. Thus, these results indicate that the combination of rapamycin and PEITC could be an effective therapeutic strategy for PTEN-deficient prostate cancer or prostate cancer in which Akt is hyperactivated.

To further address the feasibility of PEITC and rapamycin treatment for PTEN-deficient prostate cancer, we employed a mouse model for prostate cancer in which prostate Pten is specifically deleted by Cre recombinase driven by the probasin promoter (*Pbsn-Cre4;Pten*$^{f/f}$ mice). Mice that are deficient for PTEN in the prostate display progressive forms of prostatic cancer that histologically resemble human prostate cancer, ranging from mild prostatic intraepithelial neoplasia (PIN) at 10 weeks of age to large multinodular malignant adenocarcinoma with metastasis within 8 months (*Trotman et al., 2003*). Pten deletion leads to Akt activation in the prostate and, similar to in vitro observations, an increase in oxidative stress, as measured by the increased level of 4-hydroxy-nonenal (4HNE) protein adducts (*Figure 3A*). As the onset of PIN occurs within 2 months and invasive CaP occurs within 8 months, it was possible to test the efficacy of our therapeutic approach at two different stages of prostate cancer, low-grade PINs and, later, high-grade PINs and CaP stages. The first strategy is depicted in *Figure 3B*. The treatment did not significantly affect body weights of the mice (*Figure 3—figure supplement 1*), and prostate weights did not significantly change in the control mice after treatment with rapamycin alone, PEITC alone, or rapamycin and PEITC in combination (*Figure 3—figure supplement 2*). However, these treatments significantly decreased prostate weights in the *Pbsn-Cre4;Pten*$^{f/f}$ mice, which was most profound whenrapamycin and PEITC were combined (*Figure 3C*). When tumor sections were analyzed after 8 months, we found that all treatments markedly inhibited proliferation, as measured by BrdU incorporation (*Figure 3D and E*), but PEITC also induced cell death, which was further exacerbated when PEITC was combined with rapamycin (*Figure 3D and F*). Finally, the combination of rapamycin and PEITC treatment markedly increased survival (*Figure 3G*). Histopathological analysis showed that while two-thirds and one-third of untreated mice had high grade PIN and microinvasive carcinoma, respectively, one-third of mice treated with rapamycin and PEITC did not have any detectable PIN, 16% had low grade PIN and only one-third had high grade PIN and 16% microinvasive carcinoma (*Table 1*, and *Figure 3—figure supplement 3*). By contrast, treating the mice with NAC to decrease the ROS levels markedly increased the prostate weights and tumor growth (*Figure 3H*). All NAC-treated mice had carcinoma, with the majority of mice (75%) displaying invasive carcinoma and 25% microinvasive carcinoma (*Table 1* and *Figure 3—figure supplement 3*). The results indicate that high ROS levels are an impediment to tumor progression. Next, we wanted to know whether the efficacy of such a treatment was greater if the mice were treated at a younger age. Therefore the mice were treated at 2 months according to the protocol depicted in *Figure 4A*. One cohort of mice was sacrificed at 6 months, and another cohort of mice was left untreated for another 6 months and sacrificed at 12 months. A third cohort of mice was used to determine survival. As shown in *Figure 4B* and *Figure 4—figure supplement 1*, the treatments did not affect the body weights but significantly reduced the prostate weights of *Pbsn-Cre4;Pten*$^{f/f}$ mice at the 6-month time point. Analysis of tumor sections at 6 months again showed a marked decrease in cell proliferation and a marked increase in cell death with a combination of PEITC and rapamycin treatment (*Figure 4C–E*). Strikingly, the effect of PEITC and rapamycin was sustained even in the cohort of mice that were left untreated for another 6 months (*Figure 4F–I*). Interestingly, we found that BrdU incorporation was still decreased (*Figure 4H*), and cell death was increased (*Figure 4H*). Finally, treatment with PEITC and rapamycin profoundly increased survival, even though the treatment was stopped at 6 months of age (*Figure 4J*). Taken together, the results suggest that treatment with rapamycin and PEITC not only attenuates prostate tumor growth but also regresses tumor progression.

## HK2 expression is induced in Pten-deficient prostate cancer in an Akt-dependent manner

Hexokinases catalyze the first committed step of glucose metabolism by phosphorylating glucose. Hexokinase 2 (HK2), which is not expressed in most mammalian tissues, is markedly induced in cancer cells by different mechanisms (*Patra and Hay, 2013*; *Patra et al., 2013*). Previously, we showed that systemic deletion of HK2 in mice is well tolerated and is therapeutic for lung cancer (*Patra et al., 2013*). HK2 is also directly phosphorylated by Akt, which increases its binding to

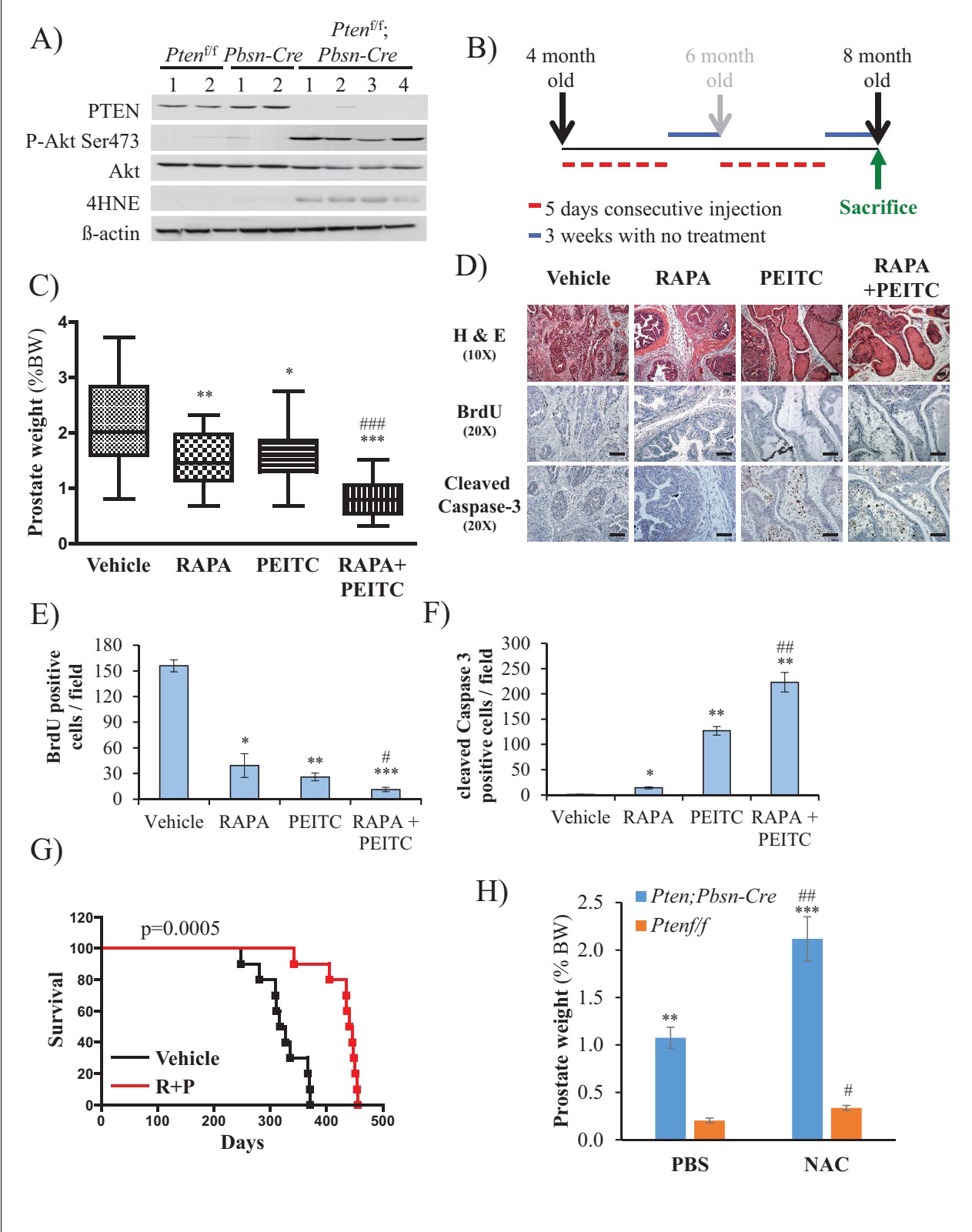

**Figure 3.** Effect of rapamycin, PEITC, and a combination of rapamycin and PEITC on cell proliferation, cell death, survival, and the tumors of *Pbsn-Cre4; Pten*^f/f^ mice. (**A**) Tissue lysates were prepared from prostates isolated from four control mice (*Pten*^f/f^ or *Pbsn-Cre*) and four *Pbsn-Cre4;Pten*^f/f^ mice. Immunoblot analysis shows the expression levels of PTEN, Akt-P (ser 473), total-Akt, p21, 4HNE, and ß-actin as a loading control. (**B**) Schematic of mouse treatment: control (*Pten*^f/f^ or *Pbsn-Cre*) and *Pbsn-Cre4;Pten*^f/f^ mice were randomly divided into four groups of 9 to 16 mice at 4 months of age, *Figure 3 continued on next page*

Figure 3 continued

and they received a daily (5 days a week) intraperitoneal injection of drugs, PEITC (35 mg/kg BW), rapamycin (2 mg/kg BW), rapamycin in combination with PEITC (1:1), or solvent control, for 6 consecutive weeks. Treatment was then interrupted for 3 weeks and resumed at 6 months of age for another 6 weeks. The mice were sacrificed at 8 months of age and examined for the presence of prostate hyperplasia. (C) Graphs showing the relative prostate weight to total body weight (% body weight) of *Pbsn-Cre4;Pten*^f/f^ mice treated with vehicle (n = 15 mice), rapamycin (RAPA, n = 11), PEITC (n = 9), or RAPA +PEITC (n = 16). The box plots represent the 25^th^ to 75^th^ percentiles (boxes) with the median, and the whiskers represent the maximum and minimum values. *p=0.05, **p=002, ***p<0.0001 versus vehicle. ###p<0.0005 versus PEITC. (D) Cross-sections of prostate tissues collected at 8 months from *Pbsn-Cre4;Pten*^f/f^ mice treated with different drugs were subjected to H and E staining (top), BrdU staining (middle), and anti-cleaved caspase-3 staining (bottom). Scale bars: 100 µm (E, F) Histograms showing quantification of the positively stained cell cross-sections shown in *Figure 3D* for BrdU (E) and cleaved caspase-3 (F). The results are presented as the mean ±SEM of positively stained cells of four sections from four treated mice. The stained cells were counted in four random fields of each section. *p<0.002, **p<0.005, ***p<0.0002 versus vehicle. #p=0.04, ##p=0.01 versus PEITC. (G) A cohort of 20 *Pbsn-Cre4;Pten*^f/f^ mice treated with vehicle (n = 10) or rapamycin in combination with PEITC (R + P; n = 10) were kept alive, and Kaplan-Meier curves of the percentage of mice survival are shown. The vehicle-treated mice have a medium survival age of 322 days versus 443 days for the 'R + P' treated mice. The p-values and median survival were calculated by log-rank tests. (H) Graph showing the relative prostate weights of *Pbsn-Cre4; Pten*^f/f^ mice (n = 15) treated with N-acetyl-cysteine (NAC, n = 9) or PBS (n = 6) at 8 months of age and 11 *Pten*^f/f^ mice (NAC, n = 7 and PBS, n = 4). The data represent the mean ±SEM. **p=0.0006, ***p=0.0001 versus *Pten*^f/f^. #p=0.01, ##p=0.003 versus PBS for each mice genotype.

DOI: https://doi.org/10.7554/eLife.32213.024

The following figure supplements are available for figure 3:

**Figure supplement 1.** Graphs showing the body weights of control (left) and *Pbsn-Cre4;Pten*^f/f^ (right) mice at the end-point (8 months).
DOI: https://doi.org/10.7554/eLife.32213.025
**Figure supplement 2.** Graphs showing the relative prostate weights of the control mice sacrificed at 8 months.
DOI: https://doi.org/10.7554/eLife.32213.026
**Figure supplement 3.** Representative histopathological images.
DOI: https://doi.org/10.7554/eLife.32213.027

mitochondria (*Miyamoto et al., 2008*), and therefore its activity (*DeWaal et al., 2018*). We therefore examined the human prostate cancer cell lines DU145, PC3, and LNCaP for expression of HK2 and found that the PTEN-deficient PC3 and LNCaP cells expressed higher levels of HK2 compared with the PTEN-proficient DU145 cells (*Figure 5A* and *Figure 5—figure supplement 1*). The high levels of HK2 in the PC3 and LNCaP cells were dependent on Akt because treatment with the pan-Akt inhibitor MK2206 diminished HK2 expression (*Figure 5A*) and knockdown of Akt1 and Akt2 in PC3 cells decreased HK2 expression (*Figure 5—figure supplement 2*). In addition, knockdown of PTEN in DU145 cells increased HK2 expression whereas expression of PTEN in PC3 and LNCaP cells decreased HK2 expression (*Figure 5—figure supplement 3*). The knockdown of HK2 only modestly decreased the total hexokinase activity in DU145 cells, while in PC3 and LNCaP cells, HK2 knockdown decreased most of the total hexokinase activity (*Figure 5B and C*). The results suggest that in the PTEN-deficient PC3 and LNCaP cells, HK2 is the major contributor of hexokinase activity. Indeed the knockdown of hexokinase 1 (HK1) in PC3 cells had only a modest effect on the total hexokinase

**Table 1.** Histopathologic analysis relative to *Figure 3*.

| | Grade | | | | |
|---|---|---|---|---|---|
| | No PIN | Low grade PIN | High grade PIN | Microinvasive carcinoma | Invasive carcinoma |
| *Pbsn-Cre4;Pten*^f/f*^ | | | 66% | 33% | |
| *Pbsn-Cre4;Pten*^f/f^ R + P^†^ | 33% | 16% | 33% | 16% | |
| *Pbsn-Cre4;Pten*^f/f^ + NAC^‡^ | | | | 25% | 75% |

*The anterior lobes of prostates from untreated mice were analyzed by histopathology at 8 months (percentage of mice with highest grade is indicated).

†The anterior lobes of prostates from mice treated at 4 months with rapamycin and PEITC (R + P) were analyzed by histopathology at 8 months (percentage of mice with highest grade is indicated).

‡The anterior lobes of prostates from mice treated at 4 months with NAC were analyzed by histopathology at 8 months (percentage of mice with highest grade is indicated).

DOI: https://doi.org/10.7554/eLife.32213.030

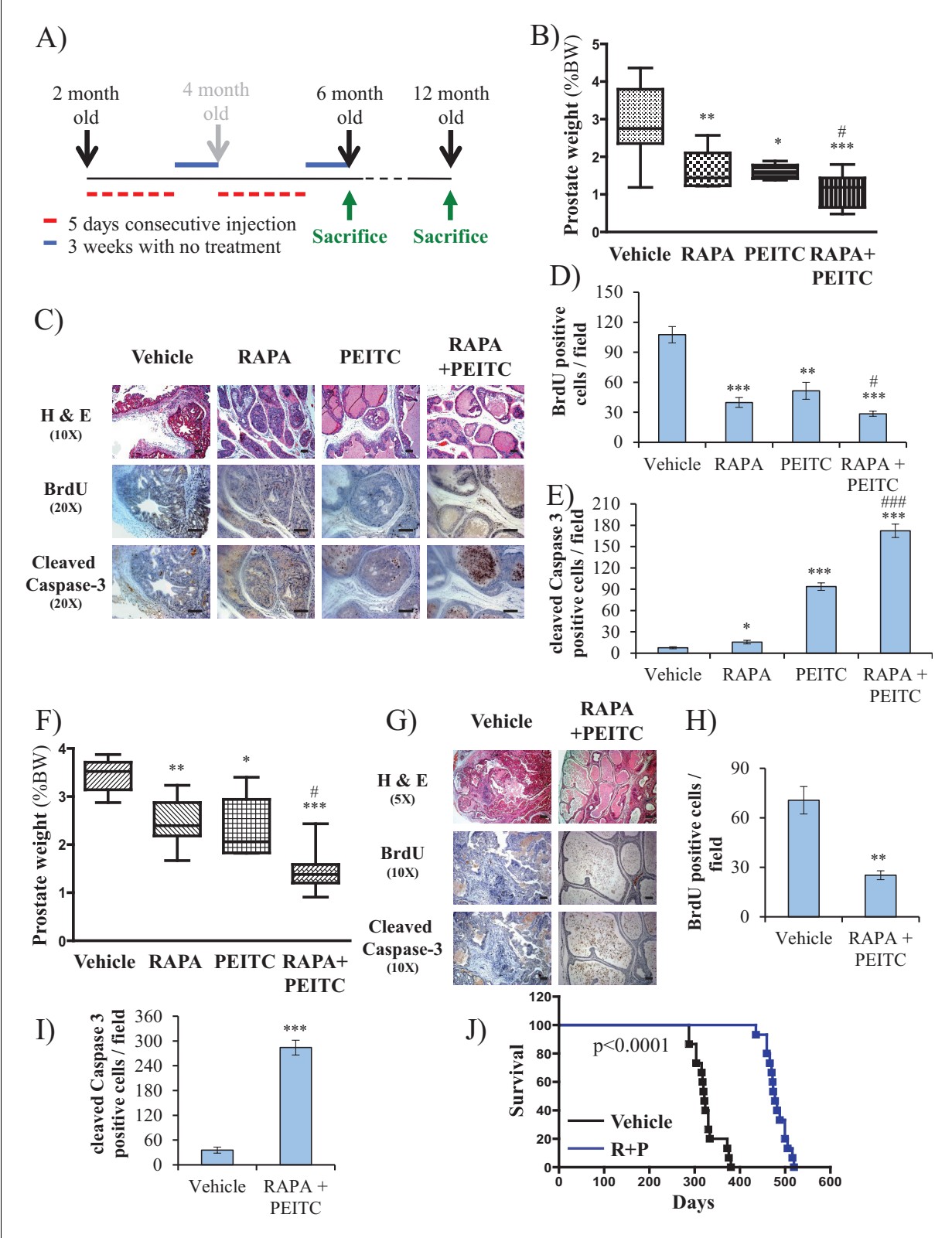

**Figure 4.** Early treatment of *Pbsn-Cre4;Pten^f/f* mice with rapamycin +PEITC inhibits tumor growth and increases survival, even after treatment was halted for 6 months. (**A**) Schematic of mice treatment: control and *Pbsn-Cre4;Pten^f/f* mice were randomly divided into four groups of four to 10 mice at 2 months of age, and they received IP drug injections as indicated in the schematic. A pool of mice was sacrificed at 6 or 12 months of age and examined for the presence of prostate hyperplasia. (**B**) Graphs showing the relative prostate weights of *Pbsn-Cre4;Pten^f/f* mice sacrificed at 6 months

*Figure 4 continued on next page*

Figure 4 continued

and treated with vehicle (n = 9), RAPA (n = 4), PEITC (n = 4), or RAPA +PEITC (n = 8). The box plots represent the 25th to 75th percentiles (boxes) with the median, and the whiskers represent the maximum and minimum values. *p=0.03, **p=0.05, ***p<0.0001 versus vehicle. #p=0.05 versus PEITC. (C) Representative cross-sections of prostate tissues were treated as described in *Figure 4A* and collected from *Pbsn-Cre4;Pten^f/f* mice treated with different drugs at 6 months. The sections were subjected to H and E staining (top), BrdU staining (middle), and anti-cleaved caspase-3 staining (bottom). Scale bars: 100 μm. (D, E) Histograms showing quantification of the positively stained cell cross-sections for BrdU (D) and cleaved caspase-3 (E). The results are presented as the mean ±SEM of the positively stained cells of four sections from four treated mice. The stained cells were counted in four random fields of each section. *p=0.03, **p<0.001, ***p<0.0001 versus vehicle. #p<0.05, ###p<0.0001 versus PEITC. (F) Graphs representing the relative prostate weights of *Pbsn-Cre4;Pten^f/f* mice sacrificed at 12 months and treated with vehicle (n = 5), RAPA (n = 7), PEITC (n = 6), or RAPA +PEITC (n = 10). The box plots represent the 25th to 75th percentiles (boxes) with the median, and the whiskers represent the maximum and minimum values. *p=0.03, **p=0.05, ***p<0.0001 versus vehicle. #p<0.05 versus PEITC. (G) Representative cross-sections of prostate tissues were treated with vehicle or RAPA +PEITC and collected at 12 months from *Pbsn-Cre4;Pten^f/f* mice left untreated for 6 months after the initial treatment. The sections were subjected to H and E staining (top), BrdU staining (middle), and anti-cleaved caspase-3 staining (bottom). Scale bars: 50 μm for 5× objective (H and E), 100 μm for 10× objective. (H, I) Histograms showing quantification of the positively stained cell cross-sections for BrdU (H) and cleaved caspase-3 (I). The results are presented as the mean ±SEM of the positively stained cells of four sections from four treated mice. The stained cells were counted in four random fields from each section. **p=0.003, ***p<0.0001 versus vehicle. (H) A cohort of 30 *Pbsn-Cre4;Pten^f/f* mice treated with vehicle (n = 15) or rapamycin in combination with PEITC (R + P; n = 15) were kept alive, and Kaplan-Meier curves of the percentage of survival of these mice is shown. The vehicle-treated mice have a median survival age of 321 days versus 477 days for the 'R + P' treated mice. The p-values and median survival for the indicated treatments were calculated by log-rank tests.

DOI: https://doi.org/10.7554/eLife.32213.028

The following figure supplement is available for figure 4:

**Figure supplement 1.** (A) Graphs showing the body weights of control (left) and *Pbsn-Cre4;Pten^f/f* (right) mice at 6 months.

DOI: https://doi.org/10.7554/eLife.32213.029

---

activity (*Figure 5—figure supplement 4*) and no effect on cell proliferation in comparison with HK2 knockdown (*Figure 5—figure supplement 5*).

## HK2 deficiency in Pten-deficient prostate cancer cells impairs proliferation and tumorigenesis and overrides chemoresistance

HK2 knockdown in PC3 and LNCaP cells markedly affected the proliferation of the cells, as measured by the cell numbers and BrdU incorporation, whereas the proliferation of the DU145 cells was not significantly affected (*Figure 5D–E*). The knockdown of HK1, however, did not affect the proliferation of PC3 cells and did not further decrease the attenuated proliferation induced by HK2 knockdown (*Figure 5—figure supplement 5*). Furthermore, the knockdown of HK2 impaired the anchorage-independent growth of PC3 cells (*Figure 5F*). PTEN-deficient prostate cancer cells are relatively resistant to etoposide because of Akt activation (*Figure 5—figure supplement 6*). However, HK2 knockdown re-sensitizes these cells to death induced by etoposide (*Figure 5G*). The inducible knockdown of HK2 in PC3 cells in nude mice after tumor onset substantially decreased tumor growth. Etoposide alone also inhibited tumor growth, although to a lesser extent. However, the combination of HK2 knockdown and etoposide prohibited tumor growth by both decreased proliferation and increased cell death (*Figure 5H* and *Figure 5—figure supplement 7*). Finally, we observed that glycolysis, as measured by ECAR, was significantly reduced in PC3 cells after HK2 knockdown as expected (*Figure 5—figure supplement 8*), but this was associated with a compensatory increase in oxygen consumption (OCR) (*Figure 5—figure supplement 9*). Consequently, the ROS levels were further increased in PC3 cells (*Figure 5—figure supplement 10*), and therefore, the cells became more sensitive to PEITC-induced cell death (*Figure 5—figure supplement 11*). These results suggest that HK2 depletion together with PEITC could be an additional therapeutic strategy for PTEN-deficient prostate cancer cells.

## Hk2 deletion in *Pbsn-Cre4;Pten^f/f* mice inhibits prostate tumor development by decreasing proliferation and increasing cell death

To further address the role of HK2 in prostate neoplasia in vivo, we crossed *Pbsn-Cre4;Pten^f/f* mice with *Hk2^f/f* mice to generate *Pbsn-Cre4;Pten^f/f;Hk2^f/f* mice. As shown in *Figure 6A*, HK2 expression was induced in the prostates of *Pbsn-Cre4;Pten^f/f* mice compared with that of the control mice. The deletion of HK2 in the *Pbsn-Cre4;Pten^f/f;Hk2^f/f* mice markedly decreased the prostate weights (*Figure 6B*) and substantially increased the survival compared with those of the *Pbsn-Cre4;Pten^f/f*

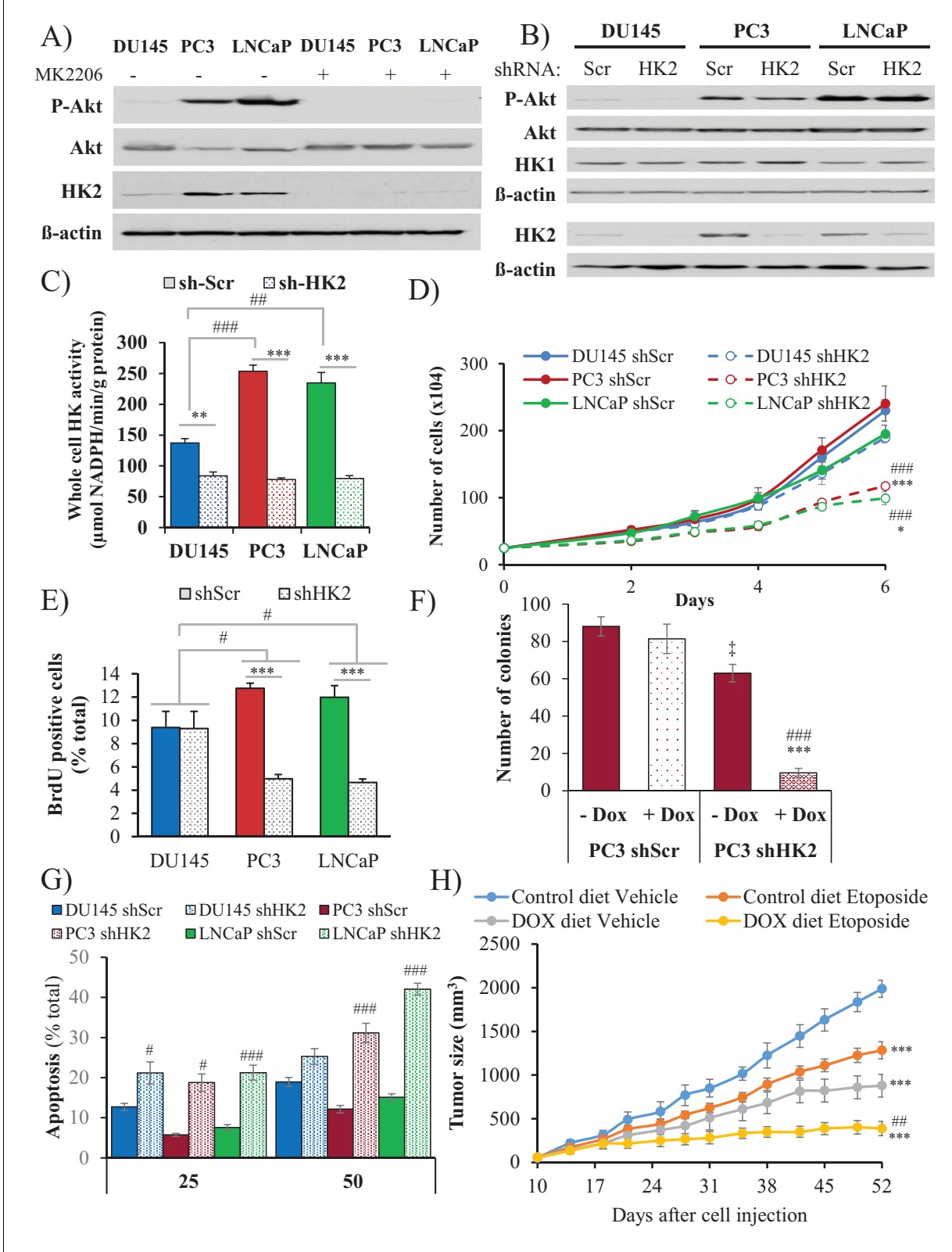

**Figure 5.** Depletion of HK2 in PTEN-deficient CaP cells inhibits proliferation, oncogenesis, and tumorigenesis while overcoming chemoresistance. (**A**) DU145, PC3, and LNCaP cells were treated with MK-2206 (0.5 μM - 24 hr) to inhibit Akt. The immunoblot shows the protein levels of P-Akt, total Akt, HK2, and ß-actin as a loading control. (**B–G**) DU145, PC3, and LNCaP cells expressing an inducible control (Scr) or HK2 shRNA were exposed to 900 ng/ml doxycycline for 5 days for HK2 deletion prior to analysis. (**B**) Immunoblot showing the protein levels of P-Akt, total Akt, HK2, HK1, and ß-actin as a

*Figure 5 continued on next page*

*Figure 5 continued*

loading control. (**C**) Graphs depicting the total hexokinase activity in these cells. The data represent the mean ±SEM of three independent experiments performed in triplicate. **p<0.002, ***p<0.001 versus shScr for each cell line. ##p<0.001, ###p<0.0001 versus DU145. (**D**) Cell proliferation after HK2 deletion in the CaP cell lines. The data represent the mean ±SEM of three independent experiments performed in triplicate. *p=0.02, ***p<0.001 versus shScr for each cell line on day 6. ###p<0.0005 versus DU145 shHK2 on day 6. (**E**) BrdU incorporation after HK2 deletion. The data represent the mean ±SEM of three independent experiments performed in triplicate. ***p<0.0001 versus shScr for each cell line. #p<0.05 versus DU145. (**F**) Anchorage-independent growth (soft-agar): PC3 Tet-ON control (SCR) and HK2-sh cells were plated in 0.35% agarose-containing medium before and after HK2 knockdown with doxycycline as described in the experimental procedures, and they were allowed to grow for approximately 3 weeks with bi-weekly media changes. The bar graphs represent the average quantification of the soft agarose colonies in PC3 cells ± SEM of three independent experiments performed in triplicate. ***p<0.0005 versus PC3 shScr +doxycycline. ‡p=0.02 versus PC3 shScr – doxycycline. ###p<0.0001 versus PC3 shHK2 – doxycycline. (**G**) After HK2 knockdown with doxycycline, cells were treated with etoposide for 24 hr before apoptosis was assessed by DAPI staining, which is presented as the percentage of apoptotic cells among total cells. The data represent the mean ±SEM of three independent experiments performed in triplicate. **p<0.001, ***p<0.0002 versus DMSO (0 μM) for each cell line. #p<0.001, ###p<0.0003 versus shScr. (**H**) In vivo therapeutic effect of etoposide in mice inoculated with PC3 prostate cancer cells. Twenty-four nude mice were injected subcutaneously with PC3 Tet-ON HK2sh cells in both flanks and randomly divided into four groups (six mice per group, 12 tumors per group) for treatment with etoposide or solvent control (Vehicle). When the tumors were palpable, two groups were provided a doxycycline diet, while the two other groups remained on the control diet. Etoposide (or vehicle) treatment was started 3 days after the diet was changed (day 13), and treatment was stopped on day 48 after tumor cell inoculation. The data represent the average size ±SEM of 12 xenograft tumors per group. Statistical analysis from day 52 (end-point): ***p<0.0001 versus the control diet vehicle. ##p<0.005 versus the doxycycline diet vehicle.
DOI: https://doi.org/10.7554/eLife.32213.031

The following figure supplements are available for figure 5:

**Figure supplement 1.** Total protein was extracted from CaP cells and subjected to immunoblotting with HK1, HK2, and ß-actin as a loading control.
DOI: https://doi.org/10.7554/eLife.32213.032
**Figure supplement 2.** Expression levels of HK2 and ß-actin as a loading control in PC3 cells in which Akt1 and Akt2 were stably knocked down.
DOI: https://doi.org/10.7554/eLife.32213.033
**Figure supplement 3.** Immunoblot showing expression of HK2 (and ß-actin as loading control) in CaP cells where PTEN is either downregulated (DU145) or overexpressed (PC3 and LNCaP).
DOI: https://doi.org/10.7554/eLife.32213.034
**Figure supplement 4.** HK1 was stably knocked down in PC3 cells after HK2 knockdown.
DOI: https://doi.org/10.7554/eLife.32213.035
**Figure supplement 5.** Cell proliferation after HK1 and/or HK2 deletion in PC3 cells.
DOI: https://doi.org/10.7554/eLife.32213.036
**Figure supplement 6.** Etoposide-induced cell death is Akt-dependent.
DOI: https://doi.org/10.7554/eLife.32213.037
**Figure supplement 7.** Data analysis for in vivo therapeutic study described in *Figure 5H*.
DOI: https://doi.org/10.7554/eLife.32213.038
**Figure supplement 8.** Effect of HK2 knockdown on ECAR.
DOI: https://doi.org/10.7554/eLife.32213.039
**Figure supplement 9.** Effect of HK2 knockdown on oxygen consumption.
DOI: https://doi.org/10.7554/eLife.32213.040
**Figure supplement 10.** Effect of HK2 knockdown on ROS levels.
DOI: https://doi.org/10.7554/eLife.32213.041
**Figure supplement 11.** Effect of HK2 knockdown on PEITC-induced cell death.
DOI: https://doi.org/10.7554/eLife.32213.042

mice (*Figure 6C*). Analysis of the prostate tumor sections showed that HK2 deletion not only inhibited tumor proliferation, as measured by BrdU incorporation, but also significantly increased apoptosis, as measured by caspase-3 cleavage *Figure 6D and E*). We concluded that HK2 is required for prostate cancer development and that its deletion induces both cytostatic and cytotoxic effects.

## Discussion

Akt is often hyperactivated in human cancers. However, systemic pan-Akt inhibition can cause toxicity and undesired effects, such as hyperinsulinemia, hyperglycemia, liver injury, and inflammation (*Wang et al., 2017*). Therefore, alternative therapeutic approaches that can selectively target cancer cells with hyperactive Akt are highly desired. Akt activation induces metabolic changes that can be exploited to selectively target cancer cells displaying hyperactive Akt. Akt is often

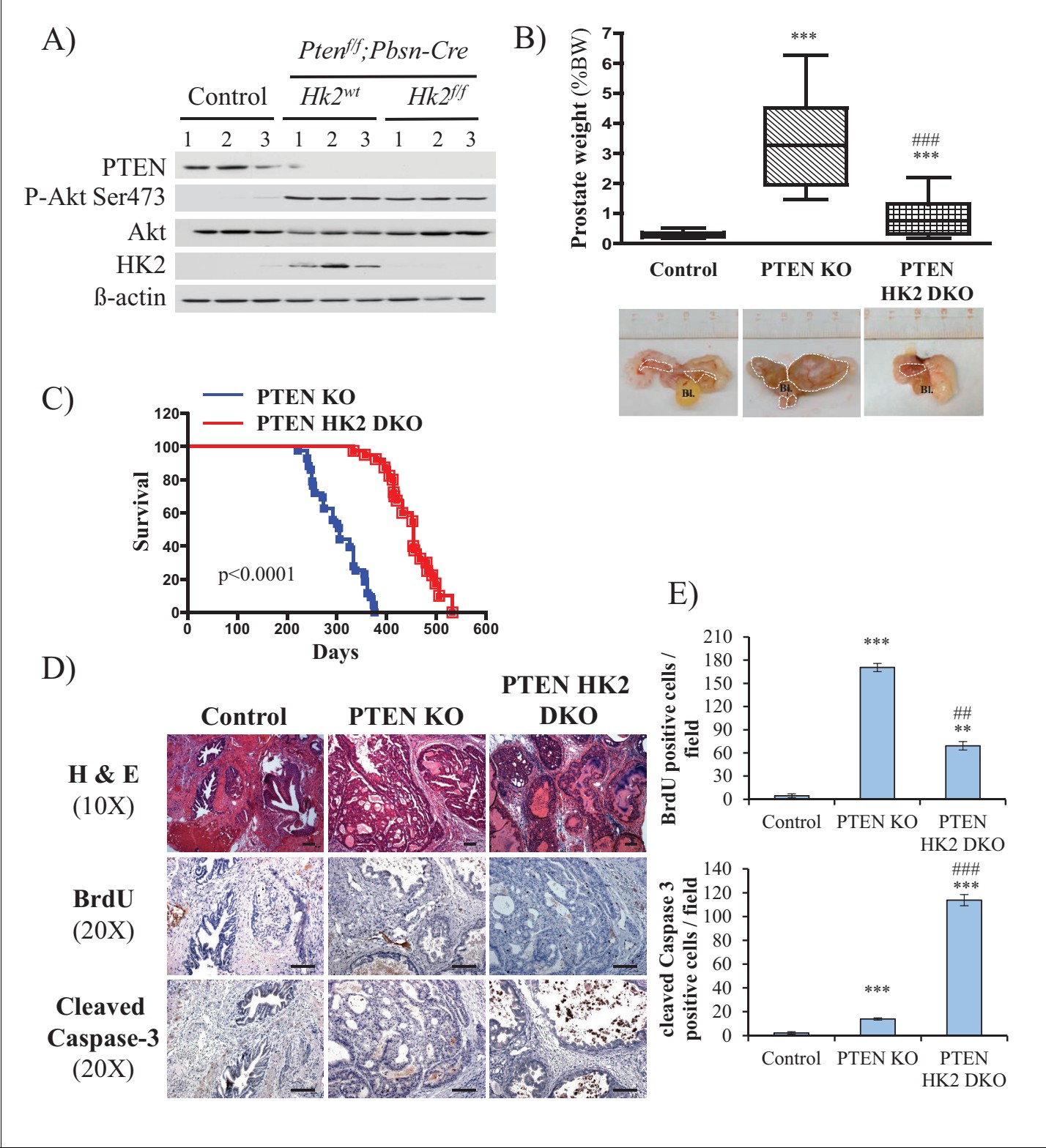

**Figure 6.** Deletion of HK2 in the prostates of *Pbsn-Cre4;Pten*[f/f] mice extends survival and inhibits tumor growth by inhibiting proliferation and increasing cell death. (**A**) Tissue lysates were prepared from prostates isolated from three control mice (*Pten*[f/f]*;HK2*[f/f]), three *Pbsn-Cre4;Pten*[f/f] mice and three *Pbsn-Cre4;Pten*[f/f]*;HK2*[f/f] mice. The immunoblot shows the expression levels of PTEN, Akt-P (ser 473), total-Akt, HK2, and ß-actin as a loading control. (**B**) Graphs showing the relative prostate weights of control (n = 23), *Pbsn-Cre4;Pten*[f/f] (PTEN KO, n = 21), and *Pbsn-Cre4;Pten*[f/f]*;HK2*[f/f] (PTEN-HK2 DKO, n = 29) mice. The box plots represent the 25th to 75th percentiles (boxes) with the median, and the whiskers represent the maximum and

*Figure 6 continued on next page*

*Figure 6 continued*

minimum values. ***p<0.0001 versus control. ### p<0.0001 versus PTEN KO. The pictures are representative of macroscopic views of the prostates (delineated by a white dash line) of control (left panel), PTEN KO (middle panel), and PTEN-HK2 DKO (right panel) mice. (C) A cohort of 43 PTEN KO and 40 PTEN-HK2 DKO mice were kept alive, and Kaplan-Meier curves of the percentage of survival of these mice are shown. The PTEN KO mice have a median survival age of 305 days versus 453 days for the PTEN HK2 DKO mice. The p-values and median survival for the indicated treatments were calculated by log-rank tests. (D) The cross-sections of prostate tissues collected at 8 months from control, PTEN KO, and PTEN-HK2 DKO mice were subjected to hematoxylin and eosin (H and E) staining (top), BrdU staining (middle), and anti-cleaved caspase-3 staining (bottom). (E) Histograms showing quantification of the positively stained cells in (D). The results are presented as the mean ±SEM of the positively stained cells of four sections from four treated mice. The stained cells were counted in four random fields of each section. **p<0.0005, ***p<0.0001 versus control. ##p<0.0005, ###p<0.0001 versus PTEN KO.

DOI: https://doi.org/10.7554/eLife.32213.043

hyperactivated in prostate cancer as a result of loss of the tumor suppressor PTEN. We therefore exploited the metabolic consequences of Akt activation in PTEN-deficient prostate cancer. Akt activation in PTEN-deficient prostate cancer elevates oxygen consumption and intracellular ROS levels. As Akt activation cannot protect cells against ROS-induced cell death, the high level of ROS mediated by Akt activation renders cells with hyperactive Akt more vulnerable to ROS-induced cell death. Rapamycin further induced Akt activity by inhibiting the feedback inhibition of Akt by mTORC1 (*Nogueira et al., 2008*). As treatment with rapamycin further increased ROS-induced cell death, we combined a ROS inducer with rapamycin as a therapeutic approach to eradicating the PTEN-deficient prostate tumors of human xenografts in mice and in a mouse model of prostate neoplasia. This therapeutic approach also converts the cytostatic effect of rapamycin to a cytotoxic effect. This strategy was successful in eradicating prostate tumors in vivo. In the mouse model of Pten-deficient prostate cancer, we found that this strategy inhibited prostate tumor growth, which was sustained even 6 months after the treatment was stopped. Interestingly, 6 months after the treatment was stopped we observed not just inhibition of proliferation, but continuous increase in cell death.

High ROS levels in cancer cells can contribute to tumorigenesis and promote pro-oncogenic signaling. However, high ROS levels also can be an impediment to tumor progression and metastasis (*Le Gal et al., 2015*; *Piskounova et al., 2015*; *Sayin et al., 2014*). Indeed, we found that in contrast to treatment with a ROS inducer, treatment with a ROS scavenger increased tumor development and invasiveness in *Pbsn-Cre; Pten^{f/f}* mice.

We found that a high level of glycolysis in PTEN-deficient prostate cancer cells is partially dependent on the ability of Akt to elevate HK2 expression. HK2 expression was not detected in the prostates of normal mice but was markedly induced after the deletion of PTEN in the prostates. In addition, HK2 is phosphorylated by Akt and increases the binding of HK2 to mitochondria (*Miyamoto et al., 2008*; *Roberts et al., 2014*). Because the binding of HK2 to mitochondria increases glycolysis (*DeWaal et al., 2018*), it is likely that Akt not only increases HK2 expression but also increases its activity in PTEN-deficient prostate cancer. HK2 knockdown in Pten-deficient prostate cancer cells in mice markedly inhibited their tumor growth and overcame their resistance to etoposide. The deletion of HK2 in the prostates of *Pbsn-Cre4;Pten^{f/f}* mice inhibited tumor growth and markedly extended their survival. Interestingly, unlike in other mouse models of cancer (*Patra et al., 2013*), HK2 deletion in the prostate of *Pbsn-Cre; Pten^{f/f}* mice is not only cytostatic but also cytotoxic.

In adult mice HK2 is not expressed in most tissues, and high expression of HK2 is limited to a small number of normal tissues (*Patra and Hay, 2013*; *Patra et al., 2013*). However, HK2 expression is markedly elevated in cancer cells. As systemic HK2 deletion is tolerated in mice, HK2 inhibition is a viable approach to circumvent chemoresistance induced by Akt activation in prostate cancer. Furthermore, it was recently demonstrated that it is feasible to develop inhibitors that preferentially inhibit HK2 and not HK1 (*Lin et al., 2016*). In summary, we provided two therapeutic approaches exploiting increased OXPHO and glycolysis levels by Akt to selectively eradicate PTEN-deficient prostate cancer.

## Materials and methods

### Cell lines

The DU145, PC3, LNCaP, 293FT, and phoenix cells were purchased from ATCC (Manassas, VA). The DU145, PC3, and LNCaP cells were maintained in RPMI-1640/10% FBS/1% pen-strep media. The 293FT and phoenix-amphotropic cells were maintained in DMEM/10% FBS/1% pen-strep media. All cells were maintained in the exponential phase of growth at 37°C in a humidified 5% $CO_2$ atmosphere. Tet-free FBS was used to maintain the Tet-ON HK2sh and Tet-ON control (shScr) cells in the absence of doxycycline, and doxycycline induction was at 900 ng/ml for the inducible DU145, PC3, and LNCaP HK2 knockdown cell lines.

All cells were confirmed to be mycoplasma negative, using the Sigma LookOut Mycoplasma PCR Detection Kit (Sigma, St. Louis, MO).

### Retrovirus and lentivirus production and infection

pBabe-Puro-PTEN-HA was previously described by Furnari *et al.* (*Furnari et al., 1997*). pBabe-Puro-mAkt was previously described by Kennedy et al. (*Kennedy et al., 1999*). Human PTEN was targeted in DU145 cells with an shRNA (5'-ACTTGAAGGCGTATACAGGA-3') cloned into the pLenti6 lentiviral vector using the BLOCK-iT Lentiviral Expression System (Thermo Fisher scientific, Waltham, MA). The sequences of the shRNAs targeting Akt1 and Akt2 to generate the PC3 Akt1/Akt2 double knockdown cells are described in (*Nogueira et al., 2008*). The sequences of the shRNAs targeting human HK2 (HK2 shRNA3) used in this study are described in (*Patra et al., 2013*). The pLKO.1 lentiviral vector containing human HK1 shRNA from Sigma was used (stock # TRCN0000037656).

Amphotropic retrovirus production was performed as previously described (*Skeen et al., 2006*). Lentiviruses were made in 293FT cells using the virapower lentiviral system (Thermo Fisher scientific) according to the manufacturer's protocol. Viruses were collected 40–50 hr after transfection, and target cells were incubated with virus for 24 hr in the presence of polybrene (8 µg/ml). Cells were selected using 9 µg/ml blasticidin, 1.3 µg/ml puromycin, or 0.2 mg/ml zeocin for 4–6 days, and a mock infection plate was used as a reference. Cells were expanded for two passages in drug-free media and frozen for subsequent use. Early passage cells were used for every experiment.

### siRNA transfection

SMARTpool ON-TARGET plus SESN3 and control non-targeting siRNA were purchased from Dharmacon (Lafayette, CO). DU145 ($8 \times 10^4$ cells/well) cells were plated in six-well plates in DMEM supplemented with 10% FBS. The next day, cells were transfected with 50 nM control-siRNA or sestrin3-siRNA using DharmaFECT reagent (Dharmacon) according to the manufacturer's instructions. Cells were split for ROS measurement or treatment with PEITC, followed by assessment of cell death 72 hr after transfection. The knockdown efficiency was analyzed by either immunoblotting or real-time PCR.

### Immunoblot analysis

For western blot analysis, $2 \times 10^6$ cells were plated on 10 cm plates and allowed to grow for 24 hr. The cells were then treated as described in the figure legends or harvested in PBS, and cell pellets were washed and frozen at −80°C. Cell extracts were then made using ice-cold lysis buffer [20 mM Hepes, 1% Triton X-100, 150 mM NaCl, 1 mM EDTA, 10 mM sodium pyrophosphate, 100 mM NaF, 5 mM iodo-acetic acid, 20 nM okadaic acid, 0.2 mM phenylmethylsulfonyl fluoride and a complete protease inhibitor cocktail tablet (Thermo Fisher)]. For the tissue extracts, frozen tissues collected by liquid nitrogen snap freezing were thawed and homogenized in the same buffer. The extracts were run on 6% to 12% SDS-PAGE gels, transferred to PVDF membranes, and probed with the following antibodies: anti-phospho-Akt Ser473, anti-panAkt, anti-cleaved caspase-3, anti-HK1, anti-HK2 anti-PTEN (Cell Signaling Technology, Danvers, MA), anti-HA (Covance, San Diego, CA), anti-4HNE (JaICA, Japan), anti-catalase, anti-CuZnSOD and anti-MnSOD (StressGen, Farmingdale, NY), anti-SESN3 (ProteinTech, Rosemont, IL), and anti-ß-actin (Sigma). Immunoblots were quantified using the NIH ImageJ software program by densitometric signal and normalized as described in figure legends.

## Cell death assays

Cells were treated as described in the figure legends, and apoptosis and cell death was quantified by DAPI staining as previously described (*Kennedy et al., 1999*) or by PI staining as previously described (*Nogueira et al., 2008*). For DAPI staining, 13% formaldehyde was added directly to the medium. After 17 hr, the media was removed and DAPI solution (1 mM in PBS) added to plates. Cells were then rinsed with PBS and visualized with immunofluorescence microscope. At least five fields per plates were scored for percentage of apoptotic cells. For quantification of apoptosis by cleaved caspase3/7 assay, cells ($15 \times 10^3$/well) were plated in a 48-well plates. Upon treatment to induce cell death, NucView-conjugated Caspase-three substrate (Nexcelom ViaStain Live Caspase 3/7 Detection) was also added at a final concentration of 4 µM. During apoptosis, caspase 3/7 proteins cleave its substrate complex and thereby release the high-affinity DNA dye (NucView), which translocates to the nucleus and binds to the DNA, producing a bright green fluorescent signal. Thirty minutes before the end of the incubation, Hoechst 33342 was added to each well (4 µg/ml) and fluorescence was measured with the Celigo Image Cytometer (Nexcelom, Lawrence, MA). The percentage of Green (apoptotic) to Total (Blue-Hoechst) is calculated.

## Measurement of ROS

Intracellular ROS generation was assessed using 2′,7′-dichlorofluorescein diacetate or dihydroethidium (Thermo Fisher scientific) as described by Nogueira et al. (*Nogueira et al., 2008*).

## NADPH and GSH assays

The intracellular levels of NADPH and total NADP (NADPH+NADP$^+$) were measured with previously described enzymatic cycling methods, as described Jeon et al. (*Jeon et al., 2012*). The intracellular levels of GSH and total glutathione (GSSG + GSH) were measured with the use of enzymatic cycling methods, as described previously (*Rahman et al., 2006*).

## Oxygen consumption assay

For the oxygen consumption measurement, two instruments were used: a Clark-type oxygen electrode and an XF96e Extracellular Flux analyzer (Agilent Seahorse). For the Clark-type oxygen electrode method, $2 \times 10^6$ cells were plated and cultured overnight. Cells were then harvested, washed with PBS, and resuspended in 500 µl of fresh RPMI. The rate of oxygen consumption was measured at 37°C using a Strathkelvin Model 782 oxygen meter equipped with a Clark-type oxygen electrode. The results are expressed as the nanomoles of oxygen consumed per minute and per million cells. For the Agilent Seahorse method, see below.

## Mitochondrial membrane potential

MMP was determined with JC-1 dye (Thermofisher) using a FACScan flow cytometer. JC-1 dye accumulates in the mitochondrial membrane in a potential-dependent manner. The high potential of the inner mitochondrial membrane facilitates formation of the dye aggregates with both excitation and emission shifted towards red light when compared with that for JC-1 monomers (green light). Cells were seeded into a 12-well black plate at a density of $10 \times 10^4$ cells/well, trypsinized and resuspended in JC-1 solution (10 µg/ml) in RPMI, and incubated in $CO_2$ incubator at 37°C for 30 min. Before measurements, the cells were centrifuged and then washed twice with the PBS and immediately analyzed by flow cytometry. Each experiment included a positive control; 10 µM of the FCCP was added to the cells as an uncoupler. Results are shown as a ratio of fluorescence measured with red to green filters (aggregates to monomer fluorescence). Each sample was run three times in triplicate.

## Measurement of the oxygen consumption rate (OCR) and the extracellular acidification rate (ECAR)

OCR and ECAR measurements were performed using the XF96e Extracellular Flux analyzer (Agilent Technologies, Santa Clara, CA). Cells were plated on XF96 cell culture plates (Agilent Technologies) at $3 \times 10^4$ cells per well. The cells were incubated for 24 hr in a humidified 37°C incubator with 5% $CO_2$ in RMPI-1640 medium (10% FBS). One hour prior to performing an assay, the growth medium in the wells of an XF cell plate was replaced by XF assay medium (XF base medium lacking

bicarbonate and HEPES containing 10 mM glucose, 1 mM sodium pyruvate, and 2 mM glutamine for OCR measurements, and 2 mM glutamine only for ECAR measurements), and the plate was transferred to a 37°C $CO_2$-free incubator. For OCR measurement, successive injection of compounds measured ATP-coupled respiration (1 µM oligomycin), maximal respiration (0.5 µM FCCP), and non-mitochondrial respiration (0.5 µM rotenone/antimycin A). Basal respiration, proton leakage, and spare respiratory capacity were then calculated using these parameters. For ECAR measurement, successive injection of compounds measured glycolysis (10 mM glucose), glycolytic capacity (1 µM oligomycin), and non-glycolytic acidification (50 mM 2-deoxyglucose). The glycolytic reserve was then calculated using these parameters. In a typical experiment, three baseline measurements were taken prior to the addition of any compound, and three response measurements were taken after the addition of each compound. The OCR and ECAR are reported as being normalized against cell counts (pmoles/min/$10^6$ cells for OCR and mpH/min/$10^6$ cells for ECAR). The baseline OCR or ECAR refers to the starting rates prior to the addition of a compound. Each experiment was performed at least three times in triplicate.

## HK activity
Whole-cell HK activity was measured as described previously (*Majewski et al., 2004*).

## Cell proliferation and BrdU incorporation
Cells ($4 \times 10^4$) were plated on 6 cm dishes in triplicate and counted every day for 6 days. Media was changed on the third day to ensure continuous natural growth. For BrdU incorporation, on the third day of proliferation, a subset of cells was pulsed with 3 µg/ml BrdU for 2 hr and fixed with 70% ethanol. In addition, immunostaining was performed with primary anti-BrdU monoclonal antibodies (Dako - Agilent) followed by a FITC-conjugated secondary antibody.

## Anchorage-independent growth assay
In brief, cells ($20 \times 10^3$) were re-suspended in a single cell suspension in 10% FBS in RPMI medium containing 0.35% agarose and plated onto a layer of 0.7% low-melt agarose-containing medium in a six-well dish. Cells were grown for 3 weeks in media, and doxycycline was replaced every 3 days. Soft-agar colonies from the entire well were counted after 3 weeks. The experiments were performed three times in triplicate.

## Real-Time PCR and primers
Total RNA was extracted using TRIzol reagent (Invitrogen), and first strand cDNA was produced with SuperScript III reverse transcriptase (Invitrogen) following the standard protocol. Quantitative PCR was performed with BIO-RAD iQ-SYBR green super-mix and the related system. Samples were assayed in triplicate, and data were normalized to the actin mRNA levels. The primer sequences for hSesn3 were 5'- ATG CTT TGG CAA GCT TTG TT −3' and 5'- GCA AGA TCA CAA ACG CAG AA −3', and the primer sequences for hActin were 5'-CCA TCA TGA AGT GTG ACG TGG −3' and 5'-GTC CGC CTA GAA GCA TTT GCG −3'.

## Mice strains and husbandry
All mice in this study were from C57BL/6 background. The *Pbsn-Cre4;Pten$^{f/f}$* mice have been described previously. *Pbsn-Cre4;Pten$^{f/+}$* mice were intercrossed to generate the following genotypes for experiments: *Pten$^{f/f}$*, *Pbsn-Cre4*, *Pten$^{f/f}$*and *Pbsn-Cre4;Pten$^{f/f}$*; *Hk2$^{f/f/}$* were described in (*Patra et al., 2013*). *Hk2$^{f/f}$* and *Pbsn-Cre4;Pten$^{f/f}$* mice were intercrossed to obtain mice with the following genotypes: *Pbsn-Cre4;Pten$^{f/f}$;Hk2$^{f/f}$* and *Pten$^{f/f}$;Hk2$^{f/f}$* which were used for experiments. All animal experiments were approved by the University of Illinois at Chicago institutional animal care and use committee.

## Xenograft studies
Male athymic mice (6 to 8 weeks old) were purchased from Charles River Laboratories and maintained in accordance with the NIH Guide for the Care and Use of Laboratory Animals. Cells (PC3 or DU145, $2 \times 10^6$/0.1 ml PBS) were injected subcutaneously into both the left and right flanks of each mouse. The mice were equally randomized into different treatment groups (see the figure legend).

When the tumors reached a size of 10 to 15 mm$^3$, the animals were treated with the indicated drugs (35 mg/kg PEITC, 2 mg/kg rapamycin, and a combination of rapamycin/PEITC (1:1)) from Monday through Friday by intraperitoneal injection. All the drugs were dissolved in solvent containing ethanol, cremophor-EL (Sigma), and PBS (1:1:8 vol ratio). Control mice were injected with an equal volume of solvent as a control. The body weights and tumor sizes of the mice were measured and recorded twice per week for the duration of the experiment. When the tumor sizes reached the endpoint criterion (e.g., a diameter greater than 2 cm), the mice were euthanized, and xenograft tumors were collected. Tumor tissues from representative mice from each group were sectioned, embedded in paraffin, and stained.

For the doxycycline-inducible experiments, PC3 Tet-ON HK2sh cells ($2 \times 10^6$ in 0.1 ml of PBS) expressing doxycycline-inducible shRNA constructs were subcutaneously injected into male nude mice. Once tumors were palpable, the mice were randomly assigned into different groups and fed regular chow (control) or doxycycline chow (200 mg/kg of diet (Bio_Serv, Flemington, NJ)), and they received an IP injection of the vehicle solvent etoposide (10 mg/kg) as described above.

## Prostate tumor development and survival curves

Control and *Pbsn-Cre4;Pten$^{f/f}$* mice were treated with vehicle, rapamycin, PEITC, or a combination of rapamycin/PEITC at the same doses described above at two different ages, 2 and 4 months. A schematic and the frequency of treatment are described in the figure legends. At the end of the study, prostate tissues were collected for immunoblot analysis (snap-freezing in liquid nitrogen) or histopathology (formalin fixation).

For the NAC study, a subset of 4-month-old control and *Pbsn-Cre4;Pten$^{f/f}$* mice received a daily (5 days a week) intraperitoneal injection of N-acetyl-cysteine (200 mg/kg, pH 7.4 in PBS) or PBS for 12 consecutive weeks. At the end of the study, tissues will be collected for immunoblot analysis (snap-freezing in liquid nitrogen) or histopathology (formalin fixation). For the survival curve experiments, the mice were monitored until their death or until humane end-point criteria was attained (e. g., distended abdomens).

## Histopathology and immunohistochemistry

Xenograft tumors (nude mice) and prostate tissues were collected at the indicated time points, rinsed in PBS, and quickly fixed in 10% formalin overnight before being subsequently preserved with 70% ethanol. The fixed tissues were then processed and embedded in paraffin. The paraffin-embedded tissues were processed, and 5 µM slides were prepared for hematoxylin and eosin (H and E) staining or immunostaining. For antigen retrieval, tissue sections were incubated at 95°C in 10 mM citric acid (pH 6.0) for 30 min. Detection was achieved using ABC-DAB kits (Vector Laboratories, Burlingame, CA), an anti-BrdU mouse monoclonal antibody (Dako# M0744), and an anti-cleaved caspase-3 (Asp175) antibody (Cell Signaling). For quantification, cells were counted from four section fields at a 40x magnification using four mice per condition.

## BrdU incorporation in mice

For the BrdU labeling experiments, mice were injected intraperitoneally with BrdU (Sigma) in PBS (0.5 mg BrdU/10 g of body weight) 2 hr prior to sacrifice and tissue collection. Tumors were collected and processed as described above. After dewaxing and rehydration, paraffin sections were digested by pepsin followed by EcoRI and Exonuclease III. The slides were then incubated with anti-BrdU and processed for immunohistochemistry as described above.

## Statistical analysis

Statistical analysis was performed using unpaired Student's t-tests. Survival curves were analyzed by log-rank tests, and the data are expressed as the mean ± SEM as indicated in the figure legends. Unless otherwise indicated, all the experiments were performed at least three times in triplicate.

## Acknowledgements

This work was supported by the ACS-IL grant 09–30 to VN, the NIH grants R01AG016927, R01 CA090764, and R01 CA206167, and the VA merit award BX000733 to NH.

## Additional information

### Funding

| Funder | Grant reference number | Author |
| --- | --- | --- |
| American Cancer Society | 09–30 | Veronique Nogueira |
| National Institutes of Health | R01AG016927 | Nissim Hay |
| U.S. Department of Veterans Affairs | BX000733 | Nissim Hay |
| National Institutes of Health | R01 CA090764 | Nissim Hay |
| National Institutes of Health | R01 CA206167 | Nissim Hay |

The funders had no role in study design, data collection and interpretation, or the decision to submit the work for publication.

### Author contributions

Veronique Nogueira, Conceptualization, Data curation, Formal analysis, Funding acquisition, Validation, Investigation, Visualization, Methodology, Writing—original draft; Krushna C Patra, Conceptualization, Resources, Methodology; Nissim Hay, Conceptualization, Supervision, Funding acquisition, Investigation, Project administration, Writing—review and editing

### Author ORCIDs

Veronique Nogueira (iD) http://orcid.org/0000-0003-3289-1760
Nissim Hay (iD) http://orcid.org/0000-0002-6245-3000

### Ethics

Animal experimentation: Animal experimentation: Mice were maintained and handled in accordance with the Animal Care Policies of the University of Illinois at Chicago and studies were approved by Animal Care and Use Committee under protocol numbers 13-036 and 14-112.

### Decision letter and Author response

Decision letter https://doi.org/10.7554/eLife.32213.049
Author response https://doi.org/10.7554/eLife.32213.050

## Additional files

### Supplementary files

• Transparent reporting form
DOI: https://doi.org/10.7554/eLife.32213.044

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
