## [Decision Letter]

[Editors’ note: this article was originally rejected after discussions between the reviewers and a plan of action to address the revisions requested, but the authors were invited to resubmit after an appeal against the decision.]

Thank you for submitting your work entitled "Selective and dramatic eradication of cancer by exploiting the metabolic consequences of Akt activation" for consideration by *eLife*. Your article has been reviewed by four peer reviewers, one of whom is a member of our Board of Reviewing Editors, and the evaluation has been overseen by a Senior Editor. The reviewers have opted to remain anonymous.

Our decision has been reached after consultation between the reviewers. Based on these discussions and the individual reviews below, we regret to inform you that your work will not be considered further for publication in *eLife*. The editors and referees judge that the manuscript will require additional experiments beyond those that you propose to provide in response to the reviewer comments. There was a sense among the reviewers that in your response you were largely proposing to address the concerns by text edits rather than by providing new data, and that this would not adequately address the issues or extend the work beyond what has previously been published. We should also note that *eLife* tends to reject papers if it would take more than 2 months to address the reviewer comments, and throughout this process there has been uncertainty whether you would be able to provide enough data in such a time period to satisfy most of the reviewers. We are returning your submission to you now in case you wish to submit elsewhere for speedy publication. If you are able to address the reviewer comments points in a way that includes new experiments and wish to resubmit your work to *eLife*, we would still be willing to consider a revised paper. Please note that it would be treated as a new submission with no guarantees of acceptance.

*Reviewer #1:*

Nogueira et al. extend findings from their 2008 publication to further support the model that activation of Akt downstream of Pten loss in prostate cancer causes high ROS. The current paper suggests some therapeutic strategies to take advantage of this, and provide some impressive data for PEITC and rapamycin causing tumor regression in a xenograft model of prostate cancer. They also present some data from a Pten-null mouse model in support of this combination, and include two figures at the end to build on their past work around HK2, and suggest HK2 is another target in prostate cancer.

The most novel and exciting data surround PEITC and rapamycin in the prostate xenograft models. Their remain some questions around how to interpret the data in the Pten-null model, and the final two figures on HK2 feel like a bit of a departure from the rest of the paper. I wonder if those data might be better served in a separate publication.

Specific comments

1) The authors use PEITC, which can deplete intracellular glutathione. This may have consequences beyond increasing ROS, although some data to confirm that it acts through glutathione depletion in their system would add to the paper. Alternatively, they might try an alternative glutathione reducing agent such as BSO, which has been in clinical trials in patients.

2) A cDNA rescue experiment to control fro shRNA off target effects should be included for key data using the Akt_1/2_ hairpins.

3) Some experiments are shown with Akt knockdown and Akt expression to confirm dependence on some phenotypes, but extending the Akt gain of function studies in the Pten wild-type cells to confirm key claims would also enhance the paper. Most importantly, showing that Akt expression in DU145 could sensitize the cells to PEITC and rapamycin would go along way toward supporting the model that active Akt is the determinant of this phenotype.

4) I am somewhat confused by the data in the Pten-null model. First, they start treating mice at 2 months and 4 months. The timing of PIN versus cancer in their hands should be confirmed, but based on the literature only invasive cancer is not expected at either time point. Thus, this is really more of a cancer prevention study. Second, how was the schedule and timing of treatment determined? They treat the animals for a very long time, and while differences are noted, they clearly have not prevented the cancer even with very long treatment since some proliferation is still observed even in the treated prostates at this late time point. This is at odds with the regressions observed in the xenografts, and perhaps indicates the combination is less effective in early stage lesions. Could treatment be initiated in older mice with invasive cancers to better match the xenograft experiment?

5) More detail is needed on the age of the mice and length of treatment when they show NAC increases prostate cancer progression. Perhaps I missed it, but it is unclear whether this timing paralleled the treatments as implied in the text.

6) The argument that finding isoform selective inhibitors for Akt would be challenging is noted. One might expect a similar challenge to generate isoform selective inhibitors of HK2, yet the conclusion of the second story in the paper is that selective targeting of HK2 might be an alternative strategy to treat Akt active prostate cancer. While the data showing HK2 is required are fine, they are less impressive than the PEITC/Rapamycin data and feel very much like a separate story. This might be better served as a short communication elsewhere.

*Reviewer #2:*

In the present manuscript Hay and colleagues propose to exploit the vulnerability generated by Akt-mediated ROS production as a therapeutic target in prostate cancer.

This mechanism of sensitization to oxidative stress is not new in cancer and the authors themselves contributed to elucidate it in their previous work (Nogueira et al., 2008). However, the link proposed in the present manuscript between Akt-induced ROS production and PTEN loss, an acknowledged driver in prostate cancer, is intriguing and potentially relevant for the clinics.

Unfortunately the manuscript lacks coherence, as it proposes two distinct parts (ROS inducer + rapamycin treatment and HK2 deficiency) that are barely integrated in the text.

Specific Comments:

1) The method used in Figure 2A-D for detection of apoptosis does not allow for the discrimination between apoptosis and other means of cell death. The authors correctly make use of caspase-3 cleavage as an apoptotic marker in vivo throughout the manuscript, but in order to make the same claim in their in vitro model they should resort to more specific methods for apoptosis detection (e.g. Annexin V staining). A brief overview of the method should be given instead along with the reference.

2) Experiments shown in Figures S1F-G, 1F and 2C are conducted only in PC3 cells. As they are key to prove the authors' point, they should be reproduced also in LnCAP cells.

3) When rapamycin treatment is introduced, the rationale for the use of this drug is not explained and is only very briefly mentioned only in the Discussion section. This might make the comprehension of the given results difficult for non-expert readers. The mechanism by which rapamycin induces an increase in ROS production by removing mTOR inhibition over Akt should be more clearly stated up front.

4) In DU145 cells, rapamycin does not substantially increase ROS production, whereas it does in cells with PTEN loss. This would suggest that PTEN control over Akt activity is epistatic over mTOR control. To better clarify this relationship, and to which extent mTOR inhibition is actually able to activate Akt in the different conditions, the authors should check the basal mTOR activity in the three cell lines, as well Akt activation upon rapamycin treatment both in vitro and in their mouse model. To further prove the proposed mechanism, they could also try to sensitize DU145 cells to rapamycin by knocking down PTEN.

5) The authors state "Interestingly, basal oxygen consumption in DU145 cells reached the maximum capacity of the respiratory chain, while PC3 and LNCaP cells have a larger spare capacity." However, the graphs show that for all cell lines basal respiration matches spare respiration.

6) As already mentioned, the findings on HK2 role in the induction of Akt-mediated ROS production, although interesting, seem more preliminary and do not integrate well with the previous part of the manuscript.

*Reviewer #3:*

This paper shows that hyper activation of AKT causes a ROS vulnerability in prostate cancer cells. Impressively, they demonstrate that rapamycin plus a ROS inducer markedly reduce tumor growth in PTEN-deficient prostate tumors in mouse models. Additionally the authors demonstrate that loss of PTEN in prostate cancer cells induces hexokinase 2 (HK2) expression to increase glucose metabolism. Furthermore, HK2 deficiency in mouse models of Pten-deficient prostate cancer elicited a marked inhibition of tumor development.

The paper is interesting and suitable for *eLife*.

I have a few comments that should be addressed.

Does the hyper activation of AKT lead to increase in mitochondrial membrane potential (MPP)? ROS generation is very sensitive to slight changes in MPP.

Any increase in MPP would increase ROS generation.

Does hyper activation of AKT lead to pentose phosphate pathway (PPP) dependent NADPH production or do AKT dependent cells use non-oxidative PPP? You could imagine that AKT hyper activation could lead to massive up regulation of PPP in certain cells resulting in better redox balance (the opposite of the findings in this paper). Also AKT is likely increasing lipogenesis thus decreasing NADPH levels. They might consider strategies to increase NADPH levels including AMPK activation, overexpression of IDH1 or malic enzymes see if it rescues from the detrimental effects of PIETC.

In Figure 5G the resensitization to etoposide due to HK2 deficiency. Does NAC rescue this? Is HK2 deficiency leading to decrease PPP flux and diminished antioxidant capacity? Are cells resensitized to etoposide in galactose media?

It is beyond the scope of the current study but it would be interesting to see if NAC would decrease the survival of PTEN-HK2 DKO to levels similar to PTEN single KO with NAC. This would formally demonstrate that the effects HK2 deficiency are due to redox imbalance.

*Reviewer #4:*

This manuscript is the follow up of previous work published by the authors in 2008 and 2012. It provides a low incremental gain of knowledge to the previous studies and is mostly based on correlative associations between AKT signaling, ROS production, metabolism and cytotoxicity.

1) It is my impression that the title is an overstatement that confuses the reader: Selective and dramatic eradication of cancer by exploiting the metabolic consequences of Akt activation. The evidence for a selective action on AKT hyperactive tumors is minimal, being this assumption based on the study of 3 established cell lines (it is worth noting that another major feature distinguishing these cell lines is LKB1 status, for which DU145 are KO). The use of "dramatic" in the title is by itself an overstatement. The demonstration of exploitation of metabolic consequences downstream AKT is minor, and ROS production is attributed by the authors correlatively to OXPHOS and based on previous studies to Sestrin3 regulation.

2) The authors mention the concept chemoresistance several times in the manuscript, but there is no data supporting this notion. The closer thing to chemotherapy they attempt to use is rapamycin, which would be far from that.

3) I perceive that, despite the fact that they do extensive pre-clinical assays in the PTEN KO model with ROS-inducers + Rapa, the study adds very little conceptual novelty to the field.

4) The authors present evidence on the interest of targeting HK2 in these tumor cells, but this notion is poorly connected to the rest of the study. Again, this evidence is based on previous reports. There is no causal connection of HK2 to ROS production. In addition, in the experiments with etoposide no selectiveness towards AKT activated tumors is presented. The outcome of the genetic cross of PTEN and HK2 KO could be intrinsically metabolic, and disconnected from ROS production.

[Editors’ note: what now follows is the decision letter after the authors submitted for further consideration.]

Thank you for resubmitting your work entitled "Selective eradication of cancer displaying hyperactive Akt by exploiting the metabolic consequences of Akt activation" for further consideration at *eLife*. Your revised article has been favorably evaluated by Sean Morrison (Senior editor), a Reviewing editor and three reviewers. There are a few remaining issues that need to be addressed before acceptance, as outlined below:

The manuscript is much improved. However, there are some minor points that should be addressed:

1) Results section. The rationale for using Rapa is the release of the negative-feedback on AKT. Yet, the fact that Rapa increases pAKT in this system is not shown.

2) Figure 4J/6G. How do the authors measure tumour-free survival? The Pten model does have full incidence of pre/tumoural masses by 6 months, as reported by them.

3) The statement "Next, we wanted to know whether the efficacy of such a treatment was greater if the mice were treated at a younger age, reflecting early detection in adult human males. Therefore, the mice were treated at 2 months according to the protocol depicted in Figure 4A" is incorrect. Treating these mice at 8 weeks has no resemblance with early detection, since these mice do not have cancer (barely PIN) and patients would have early or intermediate grade tumours.

4) What is the explanation for a decrease in pAKT upon treatment and release with Rapa+PEITC (S10C).

5) The quality of the IHC images should be improved (white balance, scale, and magnification).

6) There are some typos in the Abstract, most notably "Akt" is missing from: "…the display hyperactivated have high…".

---

## [Author Response]

[Editors’ note: the author responses to the first round of peer review follow.]

Reviewer #1:Nogueira et al. extend findings from their 2008 publication to further support the model that activation of Akt downstream of Pten loss in prostate cancer causes high ROS. The current paper suggests some therapeutic strategies to take advantage of this, and provide some impressive data for PEITC and rapamycin causing tumor regression in a xenograft model of prostate cancer. They also present some data from a Pten-null mouse model in support of this combination, and include two figures at the end to build on their past work around HK2, and suggest HK2 is another target in prostate cancer.The most novel and exciting data surround PEITC and rapamycin in the prostate xenograft models. Their remain some questions around how to interpret the data in the Pten-null model, and the final two figures on HK2 feel like a bit of a departure from the rest of the paper. I wonder if those data might be better served in a separate publication.

We thank the reviewer for these comments. We used the Pten-null model because only in this model, which recapitulates human prostate cancer, we could treat the mice at different stages, and most importantly to determine tumor free survival. As shown in Figure 3 and Figure 4 we markedly increased the tumor free survival (almost two-fold). The effect was cytotoxic and not only cytostatic indicating that the tumors are eradicated. This is also manifested by the sustained effect even 6 months after the treatment was stopped (Figure 4F-J). As for the HK2 experiments we strongly believe that they should be part of the paper. Elevation of glycolysis in Pten-deficient prostate cancer cells is dependent of Akt, which elevates HK2 expression and activity. Therefore, it represents another metabolic vulnerability of Akt activation.

Specific comments1) The authors use PEITC, which can deplete intracellular glutathione. This may have consequences beyond increasing ROS, although some data to confirm that it acts through glutathione depletion in their system would add to the paper. Alternatively, they might try an alternative glutathione reducing agent such as BSO, which has been in clinical trials in patients.

In the revised manuscript we showed that PEITC reduced GSH levels and reduced GSH/GSSG ratio (Figure 2—figure supplement 1 in the revised manuscript). BSO like PEITC preferentially killed the cancer cells that display hyperactive Akt (Figure 2—figure supplement 3 in the revised manuscript).

2) A cDNA rescue experiment to control fro shRNA off target effects should be included for key data using the Akt_1/2_ hairpins.

The hairpins used here are verified and established, and others and we previously used them. In the revised manuscript we show that the knockdown of Akt1 and Akt2 reduced ROS levels and cell death and these are restored when activated Akt is re-expressed (Figure 2—figure supplement 11).

3) Some experiments are shown with Akt knockdown and Akt expression to confirm dependence on some phenotypes, but extending the Akt gain of function studies in the Pten wild-type cells to confirm key claims would also enhance the paper. Most importantly, showing that Akt expression in DU145 could sensitize the cells to PEITC and rapamycin would go along way toward supporting the model that active Akt is the determinant of this phenotype.

First, in the revised manuscript we show that either the knockdown of PTEN or the expression of activated Akt in DU145 cells increased ROS level and rendered them more sensitive to PEITC and rapamycin (Figure 2—figure supplement 10 in the revised manuscript), and like in PC3 cells the knockdown of Akt1 and Akt2 in LNCaP cells rendered them more resistant to PEITC (Figure 2—figure supplement 11 in the revised manuscript).

4) I am somewhat confused by the data in the Pten-null model. First, they start treating mice at 2 months and 4 months. The timing of PIN versus cancer in their hands should be confirmed, but based on the literature only invasive cancer is not expected at either time point. Thus, this is really more of a cancer prevention study. Second, how was the schedule and timing of treatment determined? They treat the animals for a very long time, and while differences are noted, they clearly have not prevented the cancer even with very long treatment since some proliferation is still observed even in the treated prostates at this late time point. This is at odds with the regressions observed in the xenografts, and perhaps indicates the combination is less effective in early stage lesions. Could treatment be initiated in older mice with invasive cancers to better match the xenograft experiment?

We chose these two time points, because the onset of PIN and the enlargement of prostate is observed at 8-10 weeks, and because high grade PINs and the transition to carcinoma occurs at 4 months, and invasive carcinoma at 6-8 months. Therefore, we do not think that this could be considered cancer prevention. As for regression we observed a dramatic reduction in BrdU incorporation (10 fold decrease) when the mice were treated with rapamycin and PEITC (Figure 3E), which is almost the same as basal level of BrdU incorporation (Figure 6E). Notably, the same as with human, prostate enlargement is naturally occurring in aged mice (Figure 2—figure supplement 15). Moreover apoptosis was massively increased after treatment with rapamycin and PEITC (Figure 3F). Furthermore when mice treated between 2-6 months and were left without treatment for another six month we observed that the effect of PEITC and rapamycin and PEITC persists and BrdU incorporation was diminished and cell death continued to occur, and eventually the mice lived substantially longer than control untreated mice (Figure 4G-J).

5) More detail is needed on the age of the mice and length of treatment when they show NAC increases prostate cancer progression. Perhaps I missed it, but it is unclear whether this timing paralleled the treatments as implied in the text.

For NAC, mice were injected at 4 months of age for 12 weeks and prostates were analyzed for IHC and weight at the end of treatment (about 8½ months). Manuscript was corrected accordingly.

6) The argument that finding isoform selective inhibitors for Akt would be challenging is noted. One might expect a similar challenge to generate isoform selective inhibitors of HK2, yet the conclusion of the second story in the paper is that selective targeting of HK2 might be an alternative strategy to treat Akt active prostate cancer. While the data showing HK2 is required are fine, they are less impressive than the PEITC/Rapamycin data and feel very much like a separate story. This might be better served as a short communication elsewhere.

We would like to clarify that the purpose of the paper is to exploit metabolic vulnerabilities as a consequence of Akt activation to identify therapeutic approaches that selectively target cancer cells possessing active Akt. We found two metabolic vulnerabilities, which are Akt dependent. One is related to elevated OXOPHO and ROS levels and the other is related to increased glycolysis and hexokinase expression and activity. We found that in prostate cancer in which PTEN is lost HK2 expression is high and is dependent on Akt activity. This is also true in mice deficient for PTEN in the prostate. Furthermore, Akt directly phosphorylates HK2 increasing its activity and its ability to promote cell survival. Therefore we strongly believe that the HK2 data should be included in the manuscript, because this is also the first time demonstrating that HK2 deletion in a mouse model of prostate cancer markedly decreases tumor development and extends lifespan. As for isoform specificity it was recently published that it is feasible to selectivity target HK2 and not HK1 (Lin et al., 2016). This is discussed in the revised manuscript.

Reviewer #2:In the present manuscript Hay and colleagues propose to exploit the vulnerability generated by Akt-mediated ROS production as a therapeutic target in prostate cancer.This mechanism of sensitization to oxidative stress is not new in cancer and the authors themselves contributed to elucidate it in their previous work (Nogueira et al., 2008). However, the link proposed in the present manuscript between Akt-induced ROS production and PTEN loss, an acknowledged driver in prostate cancer, is intriguing and potentially relevant for the clinics.Unfortunately the manuscript lacks coherence, as it proposes two distinct parts (ROS inducer + rapamycin treatment and HK2 deficiency) that are barely integrated in the text.

In the revised a manuscript we show additional mechanisms such as increased NADP/NADPH and mitochondrial potential, which contribute to Akt– induced ROS. We also showed that ROS is an impediment to tumor progression as treatment with NAC increased tumor progression (Figure 3H and Figure 3—figure supplement 3, and Table 1).

The objectives of the paper are to exploit metabolic vulnerabilities of Akt in order to circumvent the use of Akt inhibitors and to selectively target prostate cancer cells displaying Akt. Therefore, we identified two metabolic vulnerabilities that we targeted. One is the increased OXPHO and ROS and the other is the increase in HK2 expression and activity. Please also see response to point 6 of reviewer 1.

Specific Comments:1) The method used in Figure 2A-D for detection of apoptosis does not allow for the discrimination between apoptosis and other means of cell death. The authors correctly make use of caspase-3 cleavage as an apoptotic marker in vivo throughout the manuscript, but in order to make the same claim in their in vitro model they should resort to more specific methods for apoptosis detection (e.g. Annexin V staining). A brief overview of the method should be given instead along with the reference.

As indicated by the reviewer we used cleaved caspase 3 to show apoptosis. The method that we used to quantify cell death (PI staining) is commonly used. In the tumor samples we quantified apoptosis by cleaved caspase 3. In the revised manuscript we changed the Y axis from apoptosis to cell death when π staining only was used. In some of the experiments we corroborated the data by apoptosis assays such as DAPI staining and quantification of caspases3/7 cleavage (Figure 2—figure supplements 5 and 6, and Figure 2—supplement 11C in the revised manuscript). In the revised manuscript we provided a brief overview of the methods.

2) Experiments shown in Figures S1F-G, 1F and 2C are conducted only in PC3 cells. As they are key to prove the authors' point, they should be reproduced also in LnCAP cells.

In the revised manuscript the NAC experiments were repeated in LNCaP cells (Figure 2—figure supplement 6 in the revised manuscript). We also knocked down Akt1 and Akt2 in LNCaP cells and showed that it decreased ROS and sensitivity to PEITC (Figure 2—figure supplement 11 in the revised manuscript).

3) When rapamycin treatment is introduced, the rationale for the use of this drug is not explained and is only very briefly mentioned only in the Discussion section. This might make the comprehension of the given results difficult for non-expert readers. The mechanism by which rapamycin induces an increase in ROS production by removing mTOR inhibition over Akt should be more clearly stated up front.

This was clearly stated in the revised manuscript.

4) In DU145 cells, rapamycin does not substantially increase ROS production, whereas it does in cells with PTEN loss. This would suggest that PTEN control over Akt activity is epistatic over mTOR control. To better clarify this relationship, and to which extent mTOR inhibition is actually able to activate Akt in the different conditions, the authors should check the basal mTOR activity in the three cell lines, as well Akt activation upon rapamycin treatment both *in vitro* and in their mouse model. To further prove the proposed mechanism, they could also try to sensitize DU145 cells to rapamycin by knocking down PTEN.

We would like to point out that rapamycin alone did not substantially increased ROS in all the cell lines tested and not just in DU145 (Figure 2—figure supplement 13 and 14). mTORC1 appears to be a downstream effector of Akt-mediated OXPHO. However, the effect of mTORC1 on OXPHO is mediated by eIF4E and not by S6K1, and rapamycin is not sufficient to inhibit eIF4E by 4EBP and therefore does not inhibit mTORC1-mediated OXPHO. Indeed, we showed in the manuscript that rapamycin does not inhibit OCR and OCR is only inhibited by mTOR kinase inhibitor (Figure 2—figure supplement 14). Thus the addition of PEITC converts rapamycin from being cytostatic to being cytotoxic. We will determine if Akt was activated by rapamycin (Figure 2—figure supplement 9).

5) The authors state "Interestingly, basal oxygen consumption in DU145 cells reached the maximum capacity of the respiratory chain, while PC3 and LNCaP cells have a larger spare capacity." However, the graphs show that for all cell lines basal respiration matches spare respiration.

The graph presented in Figure 1B, shows that baseline respiration match maximal respiration. However, when analyzing OCR data, basal respiration is calculated by subtracting non-mitochondrial respiration to baseline. In this case, we obtain the graph in Author response image 1:

So when we calculate the spare capacity (basal minus maximal respiration), DU145 cells do not have a spare capacity compared to PC3 and LNCaP cells.

6) As already mentioned, the findings on HK2 role in the induction of Akt-mediated ROS production, although interesting, seem more preliminary and do not integrate well with the previous part of the manuscript.

We did not claim that HK2 has a role in Akt-mediated ROS production. We found that the suppression of HK2 further increased ROS and further sensitizes to PEITC. However, because HK2 expression is induced in Akt-dependent manner in prostate cancer cells, and since it is directly phosphorylated by Akt, it is a metabolic vulnerability of Akt activation. Please also see response to comment 6 of reviewer 1.

Reviewer #3:This paper shows that hyper activation of AKT causes a ROS vulnerability in prostate cancer cells. Impressively, they demonstrate that rapamycin plus a ROS inducer markedly reduce tumor growth in PTEN-deficient prostate tumors in mouse models. Additionally the authors demonstrate that loss of PTEN in prostate cancer cells induces hexokinase 2 (HK2) expression to increase glucose metabolism. Furthermore, HK2 deficiency in mouse models of Pten-deficient prostate cancer elicited a marked inhibition of tumor development.The paper is interesting and suitable for eLife.I have a few comments that should be addressed.Does the hyper activation of AKT lead to increase in mitochondrial membrane potential (MPP)? ROS generation is very sensitive to slight changes in MPP.Any increase in MPP would increase ROS generation.

We thank the reviewer for the positive comments. In Figure 1—figure supplement 1 of the revised manuscript we show that MPP is higher in PC3 and LNCaP in comparison to DU145 cells.

Does hyper activation of AKT lead to pentose phosphate pathway (PPP) dependent NADPH production or do AKT dependent cells use non-oxidative PPP? You could imagine that AKT hyper activation could lead to massive up regulation of PPP in certain cells resulting in better redox balance (the opposite of the findings in this paper). Also AKT is likely increasing lipogenesis thus decreasing NADPH levels. They might consider strategies to increase NADPH levels including AMPK activation, overexpression of IDH1 or malic enzymes see if it rescues from the detrimental effects of PIETC.

We do not know if the oxidative PPP is elevated in cells with hyperactive Akt. In the original manuscript we showed two mechanisms by which Akt activation elevates intracellular ROS levels. In the revised manuscript we measured NADP/NADPH ratio and found that it is elevated in the PTEN-deficient cells (Figure 2—figure supplement 4 in the revised manuscript). The elevated NADP/NADPH could be either contributing to the high level of ROS or a consequence of high level of ROS as the cells’ attempt to combat high ROS level and therefore consume NADPH. Alternatively, high NADPH consumption for fatty acid synthesis in the PTEN-deficient cells contributes to the higher NADP/NADPH ratio.

In Figure 5G the resensitization to etoposide due to HK2 deficiency. Does NAC rescue this? Is HK2 deficiency leading to decrease PPP flux and diminished antioxidant capacity? Are cells resensitized to etoposide in galactose media?

We conducted the proposed experiments and found that replacement of glucose with galactose modestly resensitizes PC3 and LNCaP cells to etoposide (see Author response image 2). We did not find that the sensitivity to etoposide is diminished by NAC (See Author response image 3). We would like to clarify, however, that HK2 is important for cell survival downstream of Akt in PTEN-deficient cells. First we showed in the paper that its expression is induced by the activation of Akt. Second, it is the major expressed isoform in PTEN deficient cells. Third, Akt directly phosphorylates HK2 and increased its binding to the mitochondria to promote cell survival.

**Author response image 2. respfig2:** Cells were allowed to plate in 48-well plates (1x10^4^ cells/well) for 16h and then media was switched to media with 11mM glucose or 11mM Galactose for 24h prior to addition of DMSO or Etoposide (50μM) for 14h (caspase 3/7 assay) or 24h (PI staining on unfixed cells).

**Author response image 3. respfig3:** Cells were allowed to plate in 48-well plates (1.5X10^4^ cells/wells) for 16h and then pre-treated with NAC (100μM) for 2h prior to addition of Etoposide (50μM) for 24h. At end-point, PI and Hoechst 33342 were added to wells for 30min and plates were visualized with Celigo Image Cytometer, and cell death was calculated.

Reviewer #4:This manuscript is the follow up of previous work published by the authors in 2008 and 2012. It provides a low incremental gain of knowledge to the previous studies and is mostly based on correlative associations between AKT signaling, ROS production, metabolism and cytotoxicity.1) It is my impression that the title is an overstatement that confuses the reader: Selective and dramatic eradication of cancer by exploiting the metabolic consequences of Akt activation. The evidence for a selective action on AKT hyperactive tumors is minimal, being this assumption based on the study of 3 established cell lines (it is worth noting that another major feature distinguishing these cell lines is LKB1 status, for which DU145 are KO). The use of "dramatic" in the title is by itself an overstatement. The demonstration of exploitation of metabolic consequences downstream AKT is minor, and ROS production is attributed by the authors correlatively to OXPHOS and based on previous studies to Sestrin3 regulation.

In our current manuscript we provided additional mechanisms to by which Akt could increase ROS in Pten-deficient prostate cancer cells. We identified two metabolic consequences of Akt activation. First, Akt elevates OXPHO, increases NADP/NADPH ratio and therefore increased ROS. Second, Akt elevates the expression and activity of HK2 in Pten-deficient prostate cancer to increase glycolysis. To determine the efficacy of the therapeutic strategies we employed two genetically engineered mice that we followed for two years. To complement these studies we also employed xenografts of human cancer cells.

We have modified the title of the manuscript. We clearly showed in the manuscript that OXPHO, high ROS, and cell death are Akt dependent (see Figure 2—figure supplements 7-11 in the revised manuscript). We also showed that expression of high HK2 levels is Akt dependent (Figure 5—figure supplement 2 in the revised manuscript). Furthermore, Akt directly phosphorylates HK2 and increased its binding to mitochondria (Miyamoto et al., 2008). As we recently showed binding of HK2 to mitochondria is important for glycolysis (DeWaal et al., 2018).

2) The authors mention the concept chemoresistance several times in the manuscript, but there is no data supporting this notion. The closer thing to chemotherapy they attempt to use is rapamycin, which would be far from that.

It is well established that hyperactivation of Akt exerts chemoresistance. As an example we showed that Pten-deficient cells are relatively resistant to etoposide in Akt-dependent manner, and we could overcome this resistance by ablating HK2 (Figure 5G, H). In the revised manuscript we showed that expressing activated Akt in the DU145 cells renders them more resistant to etoposide, whereas the knockdown of Akt1 and Akt2 in LNCaP and PC3 cells render them more sensitive to etoposide (Figure 5—figure supplement 6).

3) I perceive that, despite the fact that they do extensive pre-clinical assays in the PTEN KO model with ROS-inducers + Rapa, the study adds very little conceptual novelty to the field.4) The authors present evidence on the interest of targeting HK2 in these tumor cells, but this notion is poorly connected to the rest of the study. Again, this evidence is based on previous reports. There is no causal connection of HK2 to ROS production. In addition, in the experiments with etoposide no selectiveness towards AKT activated tumors is presented. The outcome of the genetic cross of PTEN and HK2 KO could be intrinsically metabolic, and disconnected from ROS production.

We would like to clarify that the purpose of the paper is to exploit metabolic vulnerabilities as a consequence of Akt activation to identify therapeutic approaches that selectively target cancer cells possessing active Akt. We found that in prostate cancer in which PTEN is lost HK2 expression is high and is dependent on Akt activity. This is also true in mice deficient for PTEN in the prostate. Furthermore, Akt directly phosphorylates HK2 increasing its activity and its ability to promote cell survival. Finally we showed that HK2 KD increased ROS levels (Figure 5—figure supplement 10). Therefore we strongly believe that the HK2 data should be included in the manuscript, because this is also the first time demonstrating that HK2 deletion in a mouse model of prostate cancer markedly decreases tumor development and extends lifespan.

[Editors’ note: the author responses to the re-review follow.]

The manuscript is much improved. However, there are some minor points that should be addressed:1) Results section. The rationale for using Rapa is the release of the negative-feedback on AKT. Yet, the fact that Rapa increases pAKT in this system is not shown.

We added a new sup Figure 6 to show the increase in pAkt by rapamycin.

2) Figure 4J/6G. How do the authors measure tumour-free survival? The Pten model does have full incidence of pre/tumoural masses by 6 months, as reported by them.

We had to sacrifice mice because tumor mass reached to end-point in which we could no longer keep the mice. It might be correct that a better definition is survival and this was corrected in the revised manuscript.

3) The statement "Next, we wanted to know whether the efficacy of such a treatment was greater if the mice were treated at a younger age, reflecting early detection in adult human males. Therefore, the mice were treated at 2 months according to the protocol depicted in Figure 4A" is incorrect. Treating these mice at 8 weeks has no resemblance with early detection, since these mice do not have cancer (barely PIN) and patients would have early or intermediate grade tumours.

We agree with this assessment and corrected the text to “at early time point”.

4) What is the explanation for a decrease in pAKT upon treatment and release with Rapa+PEITC (S10C).

There is no Figure S10C in the original manuscript and in fact there is no immunoblot showing pAkt after Rapa+PEITC. The question stems from a statement in the text that was erroneously introduced. These statements were deleted in the revised manuscript.

5) The quality of the IHC images should be improved (white balance, scale, and magnification).

This was improved in the revised manuscript.

6) There are some typos in the Abstract, most notably "Akt" is missing from: "…the display hyperactivated have high…".

This was corrected in the revised manuscript.